# Adenosine triphosphate-activated prodrug system for on-demand bacterial inactivation and wound disinfection

Yuhao Weng[1], Huihong Chen[1], Xiaoqian Chen[1], Huilin Yang[2], Chia-Hung Chen ®[3] & Hongliang Tan ®[1] ✉

The prodrug approach has emerged as a promising solution to combat bacterial resistance and enhance treatment efficacy against bacterial infections. Here, we report an adenosine triphosphate (ATP)-activated prodrug system for on-demand treatment of bacterial infection. The prodrug system benefits from the synergistic action of zeolitic imidazolate framework-8 and polyacrylamide hydrogel microsphere, which simultaneously transports indole-3-acetic acid and horseradish peroxidase in a single carrier while preventing the premature activation of indole-3-acetic acid. The ATP-responsive characteristic of zeolitic imidazolate framework-8 allows the prodrug system to be activated by the ATP secreted by bacteria to generate reactive oxygen species (ROS), displaying exceptional broad-spectrum antimicrobial ability. Upon disruption of the bacterial membrane by ROS, the leaked intracellular ATP from dead bacteria can accelerate the activation of the prodrug system to further enhance antibacterial efficiency. In vivo experiments in a mouse model demonstrates the applicability of the prodrug system for wound disinfection with minimal side effects.

Bacterial infections are a global health threat due to their high morbidity and mortality[1,2]. The commonly used approach for bacterial infection treatment is the administration of antibiotics. However, the frequent and excessive use of antibiotics can easily cause bacterial resistance against antibiotics to reduce therapeutic efficacy[3]. Therefore, developing new antibacterial agents/approaches to combat bacterial resistance has always been a hot topic in bacterial infection treatment. Recently, the prodrug approach has been suggested as a promising solution to overcome bacterial resistance[4]. A prodrug is a molecule with low or no toxicity that is converted to its active parent drug by enzymatic or chemical transformation at special times and locations. Thus, compared with the parent drug, the prodrug usually has the distinct advantages of improved drug targeting and absorption, a prolonged retention time and reduced side effects[5].

Nevertheless, the antibacterial prodrugs available are still very limited. Moreover, current antibacterial prodrugs are mostly organic derivatives of antibiotics that are produced via elaborate designs and complicated synthesis procedures[6–8].

Indole-3-acetic acid (IAA) is a naturally occurring plant hormone that represents a typical prodrug for tumor therapy[9–11]. In the presence of horseradish peroxidase (HRP), nontoxic IAA can be oxidized to produce generate reactive oxygen species (ROS) with strong oxidation capabilities[12,13]. Since oxidative damage of biomolecules triggered by ROS is not species-specific, we envision that IAA in combination with HRP may be an ideal antibacterial prodrug to combat bacterial infection in addition to its tumor therapy effects. Nevertheless, some inherent limitations remain in the HRP/IAA prodrug system. For example, the different accumulation and release behaviors of IAA and

[1]National Engineering Research Center for Carbohydrate Synthesis/Key Lab of Fluorine and Silicon for Energy Materials and Chemistry of Ministry of Education, College of Chemistry and Chemical Engineering, Jiangxi Normal University, Nanchang 330022, P. R. China. [2]College of Life Science, Jiangxi Normal University, Nanchang 330022, P. R. China. [3]Department of Biomedical Engineering, City University of Hong Kong, Tat Chee Avenue, Kowloon, Hong Kong SAR, P. R. China. ✉e-mail: hltan@jxnu.edu.cn

HRP usually causes distinct spatiotemporal distributions in target sites, leading to limited ROS generation and reducing therapeutic efficacy. Although this issue could be solved by the simultaneous transport of IAA and HRP in a single carrier, premature activation of IAA easily occurs to cause side effects in heathy tissues. Accordingly, to achieve satisfactory therapeutic efficacy, the simultaneous transport of IAA and HRP while avoiding the premature activation of IAA is essential. Therefore, several delivery strategies have recently been reported by using porous materials as carriers, such as mesoporous silica[14] and metal-organic framework[15], and encouraging results have been achieved by placing IAA and HRP in separate locations within a single carrier. However, these prodrug systems still suffer from undesirable ROS release kinetics. In addition, their therapeutic efficacies are restricted by the low loading efficiency of IAA and/or limited HRP activity.

In this work, we fabricate an adenosine triphosphate (ATP)-activated HRP/IAA prodrug system for the on-demand treatment of bacterial infection. This is accomplished by the simultaneous encapsulation of HRP and zeolitic imidazolate framework-8 (ZIF-8) loaded with IAA (denoted as IAA@ZIF-8) in polyacrylamide (pAAm) hydrogel microspheres through droplet-based microfluidic technology, as shown in Fig. 1a. In the prodrug system (HRP&IAA@ZIF-8@pAAm, HiZP), the combination of ZIF-8 and pAAm microspheres is synergistic. The pAAm microsphere offers a biocompatible space for the coencapsulation of HRP and IAA@ZIF-8 to achieve the simultaneous transport of IAA and HRP in a single carrier, while the size selectivity of ZIF-8 enables the preloaded IAA to be physically isolated from HRP to avoid its premature activation. In addition, the pAAm microspheres are demonstrated to enhance the stability and catalytic efficiency of HRP due to their confinement effect. Moreover, high loading efficiencies of IAA and HRP are also obtained in the HiZP. Therefore, by taking advantage of the ATP-responsive characteristic of ZIF-8, HiZP can be activated by ATP to generate ROS, which results from the decomposition of the ZIF-8 framework and release of the preloaded IAA to react with the coconfined HRP in the pAAm microspheres. Since ATP is a well-known secretion of living bacteria, we highlight the functionality of HiZP as a prodrug system for bacterial inactivation (Fig. 1b). Attractively, we find that upon disruption of the bacterial membrane by ROS, the leaked intracellular ATP from dead bacteria can accelerate the activation of HiZP to further enhance antibacterial efficiency. On this basis, the applicability of HiZP for wound disinfection with negligible side effects was demonstrated by the in vivo antibacterial experiments in a mouse model.

## Results and discussion
### Responsivity of ZIF-8 to ATP
ZIF-8 was synthesized by the facile self-assembly of $Zn^{2+}$ and 2-methylimidazole (2-MeIM) through chemical coordination. The as-prepared ZIF-8 showed a nearly spherical shape with an average size of ~670 nm (Supplementary Fig. 1). To demonstrate the ATP-responsive characteristic of ZIF-8 more clearly, fluorescein (FAM) was employed as a visual indicator for encapsulation in ZIF-8 during the self-assembly process of $Zn^{2+}$ and 2-MeIM, producing FAM@ZIF-8. FAM is a commonly used fluorochrome with green fluorescence. The SEM image (Fig. 2a) shows that FAM@ZIF-8 is monodispersed and has a shape and size similar to that of pure ZIF-8, indicating that the presence of FAM does not affect the formation of ZIF-8. However, after incubation with ATP, obvious swelling and melting behaviors were observed from FAM@ZIF-8 (Fig. 2b), which is consistent with the decrease in the hydrodynamic diameter of FAM@ZIF-8 (Supplementary Fig. 2). By measuring the fluorescence of FAM@ZIF-8 (Fig. 2c), we saw that only the sediment exhibits strong green fluorescence, while no fluorescence was found in the supernatant. However, after adding ATP, strong green fluorescence appeared in the supernatant of FAM@ZIF-8, similar to that observed with free FAM. This result indicates that ZIF-8 is highly sensitive to ATP, which can destroy the structure of ZIF-8 to cause FAM leakage. Furthermore, we tested the responsive behaviors of FAM@ZIF-8 to analogs of ATP, including cytidine triphosphate (CTP), guanosine triphosphate (GTP) and uridine triphosphate (UTP). The results in Fig. 2d show that only ATP can cause remarkable green fluorescence in the supernatant of FAM@ZIF-8. The supernatants of FAM@ZIF-8 with the ATP analogs did not display fluorescence under identical conditions, reflecting that these analogs have no influence on the structure of ZIF-8. Thus, the responsivity of ZIF-8 to ATP is highly specific. This could be attributed to the higher affinity of ATP for $Zn^{2+}$ than 2-MeIM[16], which replaces 2-MeIM in ZIF-8 and results in the collapse of the ZIF-8 framework to release FAM.

By using the same procedures used to synthesize FAM@ZIF-8, IAA was encapsulated into ZIF-8 to produce IAA@ZIF-8. The as-obtained IAA@ZIF-8 (Supplementary Fig. 3) had a morphology and size

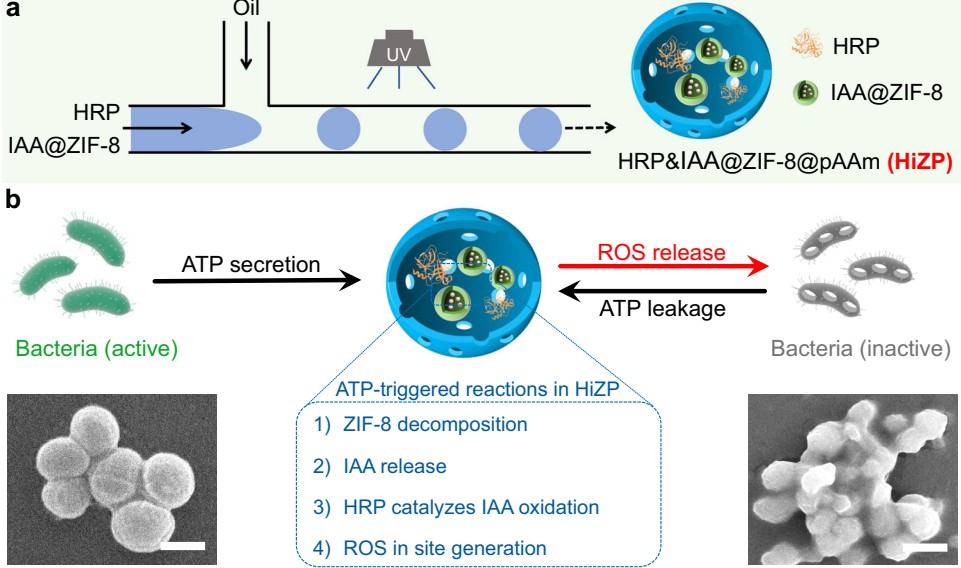

**Fig. 1 | Schematic illustration of the fabrication process and antibacterial mechanism of HiZP. a** The fabrication process of HiZP using droplet-based microfluidic technology. **b** ATP-triggered activation of HiZP for on-demand bacterial inactivation. Scale bars are 500 nm.

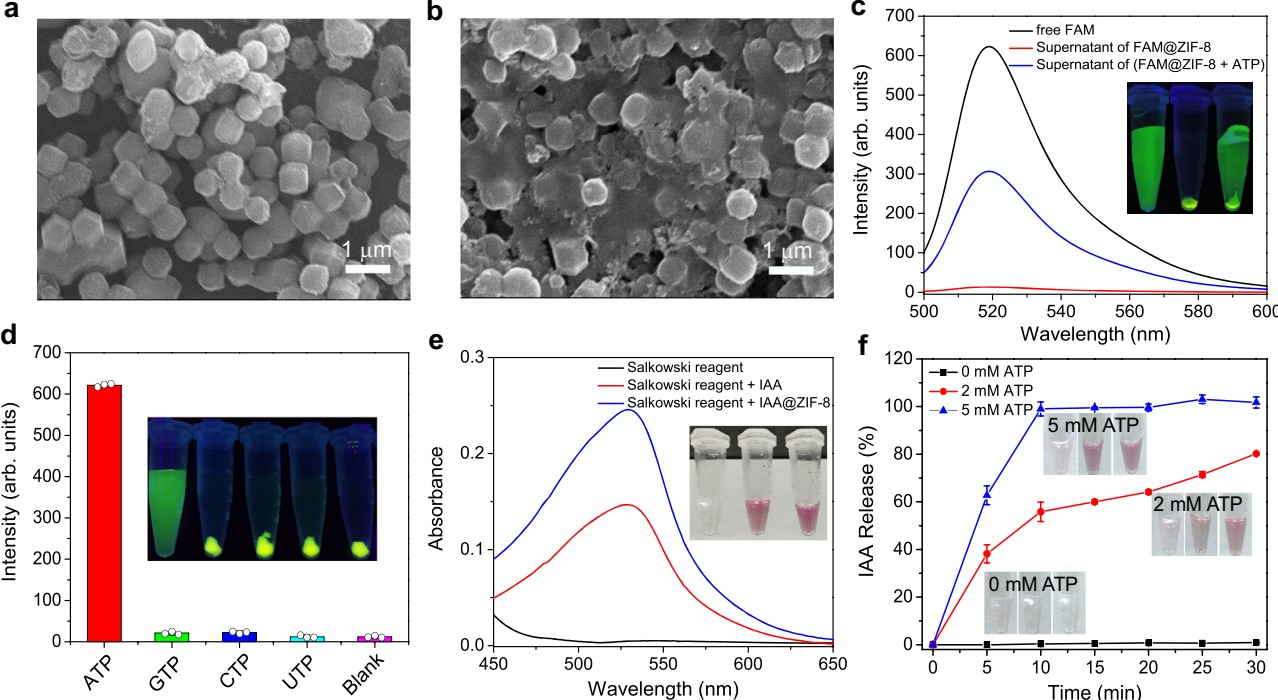

**Fig. 2 | Responsivity of ZIF-8 to ATP.** SEM images of FAM@ZIF-8 (**a**) before and (**b**) after treatment with ATP. Scale bars are 1 μm. The SEM images are representatives of three independent experiments with similar results. **c** Emission spectra of free FAM and the supernatants of FAM@ZIF-8 alone and in the presence of ATP. **d** Fluorescence responses of FAM@ZIF-8 to ATP and its analogs under identical conditions. **e** Absorption spectra of the Salkowski reagent under different conditions. **f** Time-dependent release profiles of IAA from IAA@ZIF-8 treated with different concentrations of ATP. Data are presented as mean values ± SD ($n = 3$ independent samples).

distribution similar to those of ZIF-8 and FAM@ZIF-8. To verify the encapsulation of IAA, a classic colorimetric Salkowski assay was performed. In the Salkowski assay, IAA can react with the Salkowski reagent (0.5 M FeCl$_3$ dissolved in 35% perchloric acid) to form a pink product with a maximum absorption peak at 529 nm[17]. As expected, a strong absorption peak at 529 nm was measured after the reaction of the Salkowski reagent with IAA@ZIF-8 (Fig. 2e), giving the same result as that of free IAA and confirming that IAA was successfully loaded in ZIF-8 to form IAA@ZIF-8. According to the calibration plots (Supplementary Fig. 4), the loading efficiency of IAA was determined to be 76.24%. The release profile of IAA@ZIF-8 showed that less than 5% of the loaded IAA was released from IAA@ZIF-8 within 70 h (Supplementary Fig. 5), revealing that IAA was tightly trapped in ZIF-8 with negligible leakage. This was attributed to the electrostatic interactions (Supplementary Fig. 6a) and chemical coordination interactions (Supplementary Fig. 6b) between IAA and ZIF-8. However, similar to FAM@ZIF-8, the presence of ATP caused the release of IAA from IAA@ZIF-8 (Supplementary Fig. 7). The release rate of IAA from IAA@ZIF-8 was highly dependent on the ATP concentration, as the rate increased with increasing ATP concentration (Fig. 2f). This result indicates that the ATP-responsive characteristic of ZIF-8 is independent of the presence of guest molecules (i.e., FAM or IAA).

## Fabrication and characterization of HiZP

We employed droplet-based microfluidic technology (Fig. 1a) to fabricate the HiZP due to its precise generation and repeatability capabilities[18,19]. The optical image (Fig. 3a) shows that the HiZP is a highly monodispersed microsphere with a uniform size of ~190 μm. By analyzing the structure of HiZP by XRD (Fig. 3b), we found that HiZP displays the same powder XRD pattern as pure ZIF-8, suggesting the presence of IAA@ZIF-8 in HiZP. To confirm this speculation (that IAA@ZIF-8 was loaded in HiZP), a Salkowski colorimetric assay was carried out. Similar to free IAA, the addition of HiZP led to a

characteristic Salkowski absorption spectrum (Supplementary Fig. 8). This indicates that IAA was presented in HiZP. Therefore, the observations of the characteristic XRD pattern of ZIF-8 and the Salkowski absorption spectrum from HiZP validated the successful loading of IAA@ZIF-8 into the pAAm microspheres, which is further supported by the cross-sectional SEM image of HiZP and its corresponding characteristic EDS mapping of Zn element (Supplementary Fig. 9). Indeed, compared with pure ZIF-8 and free IAA, the unchanged XRD pattern and Salkowski absorbance of HiZP substantially reflect that IAA@ZIF-8 possessed excellent structural stability during the HiZP fabrication process, which not only retained the crystal nature of ZIF-8 but also effectively trapped IAA from leaking.

To verify the loading of HRP in the HiZP, a bicinchoninic acid (BCA) assay was first performed for protein determination[20]. As shown in Fig. 3c, upon the addition of the BCA reagent, a typical protein absorption band centered at 560 nm was observed in the HiZP sample, which was also seen in the free HRP sample. In contrast, in the presence of the pAAm microspheres and IAA@ZIF-8, no measurable absorption band was displayed. This reveals that HRP was successfully encapsulated into HiZP. The encapsulation efficiency of HRP was determined to be 87.05% according to the calibration plots (Supplementary Fig. 10), which is much higher than conventional HRP hosts such as porous silica (~15%)[21]. Next, the catalytic activity of the loaded HRP in HiZP was examined by employing 2,2'-azino-bis(3-ethylbenzothiazoline-6-sulfonic acid) (ABTS) as a substrate, which can be oxidized by HRP to generate green ABTS$^{\cdot+}$ with the assistance of H$_2$O$_2$[18]. From Fig. 3d, we can see that under identical conditions, HiZP had a catalytic activity similar to that of free HRP, indicating that the HRP catalytic activity did not suffer significant alterations during the HiZP fabrication process. This result was further supported by the unchanged tryptophan emission (Supplementary Fig. 11)[22–24] and similar steady-state kinetic data ($K_m$, $V_{max}$ and $K_{cat}$ values) of HiZP compared to free HRP (Supplementary Fig. 12 and Supplementary Table 1). In addition,

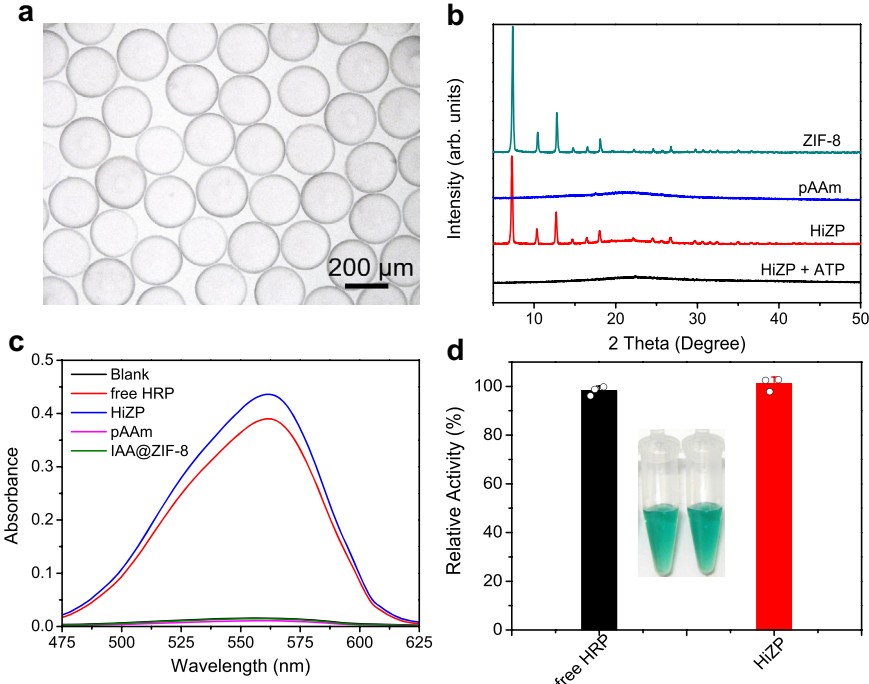

**Fig. 3 | Characterization of HiZP. a** Optical image of HiZP. The image is representative of three independent experiments with similar results. **b** Powder XRD patterns of ZIF-8, pAAm, HiZP, and HiZP treated with ATP. **c** Absorption spectra of the BCA reagent alone and in the presence of free HRP, HiZP, pAAm, and IAA@ZIF-8. **d** Relative catalytic activities of free HRP and HiZP based on the ABTS colorimetric assay. Data are presented as mean values ± SD ($n = 3$ independent samples).

by investigating the release profile of HRP from HiZP (Supplementary Fig. 13), it was found that the release rate of HRP from HiZP was approximately 20-fold lower than that from the pAAm hydrogel microsphere loaded with only HRP (i.e., HRP@pAAm). Such different release behaviors may arise from the electrostatic adsorption of HRP on the surface of IAA@ZIF-8, as they have opposite zeta potentials (Supplementary Fig. 6a), which results in the formation of a large composite to prevent the leakage of HRP from HiZP[25–27].

## ATP-triggered activation of HiZP

To examine the feasibility of using ATP to activate HiZP to generate ROS, a fluorescence assay with 2',7'-dichlorofluorescein diacetate (DCFH) was conducted. DCFH is a well-known fluorescent probe for the sensitive detection of ROS, which itself is nonfluorescent but it can be oxidized by ROS to produce highly fluorescent 2',7'-dichloro-fluorescein (DCF) (Supplementary Fig. 14)[28,29]. From Supplementary Fig. 15a, we can see that a typical DCF emission spectrum with a maximum peak at 520 nm was recorded from the DCFH solution with HRP and IAA. In contrast, the presence of HRP or IAA alone did not display any measurable DCF fluorescence under identical conditions. These data indicate that the interaction of HRP with IAA can lead to the generation of ROS to oxidize DCFH, due to the enhanced fluorescence intensity of DCF at 520 nm with increasing IAA concentration (Supplementary Fig. 15b).

Next, the effects of HiZP on the fluorescence of DCFH were investigated. As shown in Fig. 4a, HiZP itself is not capable of oxidizing DCFH. However, after the addition of the mixture of ATP and HiZP, obvious fluorescence from DCF was measured from the solution originally containing DCFH, signifying the generation of ROS from the interaction between ATP and HiZP. By examining the fluorescence of the DCF produced from DCFH under different conditions (Fig. 4b), it was found that only ATP with HiZP can cause remarkable fluorescence, while no fluorescence was detectable in the presence of HRP, IAA, $Zn^{2+}$, 2-MeIM, or pAAm. This further demonstrates that the generation of ROS originated from the interaction of HiZP with ATP, in which ATP

destroys the structure of ZIF-8 to release the preloaded IAA from IAA@ZIF-8 and thereby initiates the oxidation reaction of IAA with HRP to generate ROS. The dissociation of IAA@ZIF-8 was verified by the vanishing characteristic EDS mapping of Zn element (Supplementary Fig. 9) and XRD pattern (Fig. 3b) of ZIF-8 from HiZP.

From Supplementary Fig. 16, a time-dependent fluorescent response of HiZP to ATP can be observed. By plotting the DCF fluorescence intensity at 520 nm against ATP concentration (Fig. 4c), we can see that the DCF fluorescence gradually increased with increasing ATP concentration. This indicates that a high level of ATP can activate HiZP more effectively. Interestingly, we found that under identical conditions, the ROS yield of HiZP was approximately 4-fold higher than that of the mixture of HRP and IAA (Fig. 4d). This could be attributed to the confinement effect from the host pAAm microspheres[30–32], which brings HRP and IAA in close proximity to benefit the in situ generation of ROS.

## Antibacterial capability of HiZP

Inspired by the fact that bacteria generally secrete ATP during growth[33,34], the potential of HiZP as a prodrug system was explored in antibacterial applications. *Staphylococcus aureus* (*S. aureus*) was used as the model bacterium. Figure 5a shows that almost all bacterial cells were killed after *S. aureus* was incubated with HiZP. Compared with untreated *S. aureus* (blank), no obvious antibacterial effects were observed after incubation with the individual components of HiZP, including pAAm, HRP, IAA and ZIF-8. By recording the optical density at 600 nm ($OD_{600}$) to monitor bacterial growth, we found that HiZP treatment effectively inhibited the growth of *S. aureus*, as indicated by its low $OD_{600}$ value (Supplementary Fig. 17). However, remarkable $OD_{600}$ values were still measured after exposure to pAAm, HRP, IAA or ZIF-8, even when they were present at high concentrations. These results indicate that the individual components of HiZP possess good biocompatibility and have negligible toxicity against *S. aureus*. Thus, the observed bacterial inactivation should be attributed to the integrated effects of the components that form HiZP. Upon increasing the

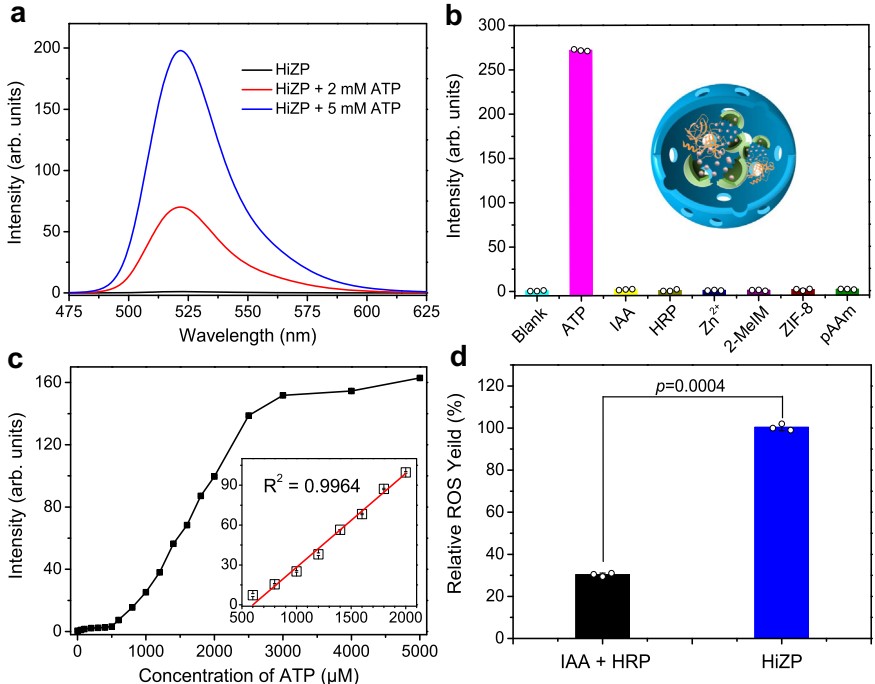

**Fig. 4 | ATP-triggered activation of HiZP. a** Emission spectra of DCFH alone and in the presence of HiZP with different concentrations of ATP. **b** DCF fluorescence at 520 nm under different conditions. **c** The changes of DCF fluorescence at 520 nm in the presence of ATP with concentration from 0 to 5 mM. Inset is the linear plots of DCF fluorescence intensity at 520 nm against ATP concentration. **d** Relative ROS yields of the free HRP/IAA system and HiZP. Data are presented as mean values ± SD (*n* = 3 independent samples). Statistical significance was analyzed via one-way ANOVA with Tukey's multiple comparisons test.

HiZP concentration, a dose-dependent decrease in *S. aureus* viability was observed (Fig. 5b), which further demonstrates the antibacterial activity of HiZP.

To discern whether the antibacterial ability of HiZP arises from the activation of ATP secretion by bacteria, we measured the amount of ATP in the culture medium containing *S. aureus* by using a commercial ATP colorimetric kit. As shown in Fig. 5c, in fresh medium (without *S. aureus*), no measurable absorption was observed, suggesting that ATP was not present in fresh medium. However, a clear absorbance band centered at 635 nm was found after the addition of culture medium containing *S. aureus*. This band was also seen in the absorption spectrum of free ATP, revealing that ATP was secreted during *S. aureus* growth. Thus, to further demonstrate that ROS production from HiZP is triggered by bacterial-secreted ATP, we inhibited ATP production in *S. aureus* by adding iodoacetic acid and subsequently monitored DCF fluorescence after incubation in culture medium with HiZP. As a well-known inhibitor of ATP generation, iodoacetic acid causes a significant decrease in the amount of extracellular ATP[35,36]. As expected, the DCF fluorescence gradually decreased with increasing iodoacetic acid concentration (Fig. 5d), confirming that the secreted ATP activated HiZP to produce ROS. Moreover, the ATP-dependent DCF fluorescence signal also suggests the potential of HiZP to act as an antibacterial agent for on-demand bacterial inactivation.

Moreover, compared with the mixture of free HRP and IAA (denoted as the free HRP/IAA system), much stronger (~ 7-fold) antibacterial activity was observed from HiZP (Fig. 5e), which was confirmed by its higher ability to inhibit colony formation on Luria-Bertani (LB) agar plates (Supplementary Fig. 18). The superior antibacterial activity of HiZP can be attributed to the higher levels of ROS produced by HiZP due to the confinement effects of the pAAm microspheres (Fig. 4d). In addition to this effect, it was also found that the host pAAm microsphere can contribute to improving the biological and chemical stability of HRP. Figure 5f shows that compared with untreated HiZP, more than 90% of the initial antibacterial activity was retained in the HiZP treated with excessive trypsin for 24 h. However, under identical

conditions, the surviving percentage from the free HRP/IAA system was greater than 85%, reflecting its weak antibacterial activity. This indicates that pAAm microspheres as hosts can protect HRP from protease degradation, thus endowing HRP with enhanced biological stability. Similarly, greatly improved storage stability was also observed for the HiZP compared to the free HRP/IAA system (Fig. 5g). Therefore, the pAAm microspheres are essential for HiZP to achieve high antibacterial activity because they not only provide a confined space for the coencapsulation of HRP and IAA@ZIF-8 to generate ROS in situ with high efficiency but also shield HRP against harsh environments to boost the practical applications of the HRP/IAA system in biological media. In addition, different from free HRP/IAA system, the HiZP is an integrated prodrug system with a positively charged surface (+2.76 mV), which enables HiZP to bind *S. aureus* with negatively charged surface (-20.33 mV) through electrostatic interactions and thereby inactivates bacteria more efficiently[37–39].

## Antibacterial mechanism of HiZP

To understand the antibacterial mechanism of HiZP, we first performed a live/dead fluorescent staining assay. From Fig. 6a, we can see that similar to untreated bacteria, the bacteria incubated with pAAm, HRP, ZIF-8 or IAA all survived, as indicated by the detection of only green fluorescence (fluorescein diacetate; FDA). However, a few red fluorescent spots from propidium iodide (PI) were observed in the presence of the free HRP/IAA system, revealing that this system can lead to bacterial inactivation. Nevertheless, under identical conditions, the proportion of dead bacteria (red fluorescence) produced by the free HRP/IAA system was much lower than that from the HiZP. This indicates that HiZP has superior antibacterial activity compared with the free HRP/IAA system, which is consistent with the results of the plate count assays and growth-inhibition assay, as demonstrated above. Indeed, the observation of red PI fluorescence considerably suggests that treatment with HiZP could result in disruption of the bacterial membrane. To verify this assumption, the integrity of the bacterial membrane was examined

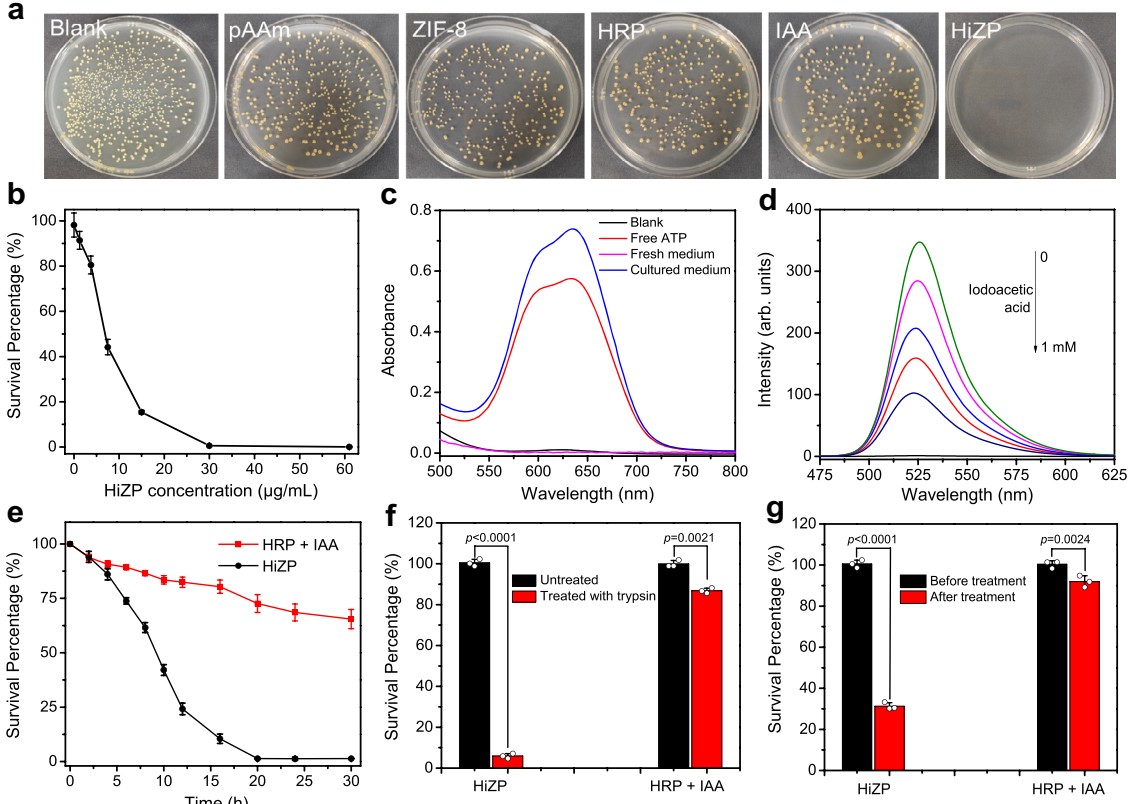

**Fig. 5 | Antibacterial capability of HiZP. a** Images of *S. aureus* colonies formed on LB agar plates after different treatments. The images are representatives of three independent experiments with similar results. **b** Viability of *S. aureus* treated with different concentrations of HiZP. (**c**) Absorption spectra produced by the addition of free ATP, fresh medium without *S. aureus* and culture medium with *S. aureus*. **d** Emission spectra of DCF in the presence of mixtures of HiZP and the supernatants of bacteria treated with different concentrations of iodoacetic acid.

**e** Time-dependent viability of *S. aureus* treated with the free HRP/IAA system and HiZP. The antibacterial activities of HiZP and the free HRP/IAA system against *S. aureus* before and after treatment with (**f**) excess trypsin and (**g**) after storage for 14 days at room temperature. Data are presented as mean values ± SD (*n* = 3 independent experiments). Statistical significance was analyzed via one-way ANOVA with Tukey's multiple comparisons test.

by SEM. From Fig. 6b, we can see that *S. aureus* treated with pAAm, HRP, ZIF-8 or IAA alone were typically spherical in shape with smooth and intact membranes, the same as those of untreated *S. aureus*. However, after treatment with the free HRP/IAA system and HiZP, the membranes of *S. aureus* were seriously deformed, and some were even destroyed. These results indicate that the antibacterial activity of HiZP is highly related to the strong oxidizing properties of ROS, which can destroy the bacterial membrane. The bacterial membrane damage was further validated by the alteration of bacterial membrane potential upon treatment with HiZP, as demonstrated by the observation of typical green fluorescence of DiBAC4(3) from the stained *S. aureus* (Fig. 6c and Supplementary Fig. 19a) and the enhanced fluorescence intensity of DiBAC4(3) upon the increase of HiZP concentration (Supplementary Fig. 19b)[40,41]. Although few green fluorescence spots were also recorded from the *S. aureus* treated with free HRP/IAA system, its fluorescence intensity and density are much lower than that of the treated *S. aureus* with HiZP. This reveals that free HRP/IAA system has a weaker effect on the bacterial membrane damage, which is consistent with its inferior antibacterial activity to HiZP.

Generally, damage to the bacterial membrane can cause the leakage of intracellular components into the surrounding environment. Since DNA and RNA have a characteristic absorption at 260 nm, the optical density value at 260 nm ($OD_{260}$) is often regarded as an indicator of bacterial membrane damage[42]. From Fig. 6d, we can see that the supernatant of the untreated bacteria shows no measurable absorbance at this wavelength. However, an

obvious absorption band centered at 260 nm was recorded from the supernatant of the bacteria treated with HiZP, which was also observed in the spectrum of pure DNA. Moreover, with increasing HiZP concentration, a dose-dependent increase in the $OD_{260}$ value was obtained (Supplementary Fig. 20). These results confirm that HiZP effectively damaged the bacterial membrane, leading to the leakage of intracellular components. Interestingly, by consecutively monitoring ATP concentration, it was found that after the addition of HiZP, a sheer rise in ATP concentration was measured in the supernatants of the treated bacteria (Fig. 6e). This reflects that once HiZP was activated by bacterial secreted-ATP, the generated ROS will disrupt bacterial membrane to cause the leakage of intracellular ATP into surrounding environment, and consequently accelerating the activation of HiZP to produce a large amount of ROS and thereby kill bacteria more efficiently.

From Fig. 6f, we found that similar to the gram-positive bacterium *S. aureus*, HiZP can also destroy the bacterial membrane of gram-negative *Escherichia coli* (*E. coli*), ultimately resulting in *E. coli* cell death. Furthermore, we examined the antibacterial activity of HiZP in drug-resistant bacterial strains by employing ampicillin-resistant *S. aureus* (amp[r] *S. aureus*) and ampicillin-resistant *E. coli* (amp[r] *E. coli*) as models. The results in Fig. 6g show that HiZP also displays high antibacterial activity against these two drug-resistant bacteria, as almost no surviving bacteria were observed after treatment with HiZP. These results clearly demonstrate that HiZP possesses broad-spectrum antibacterial ability.

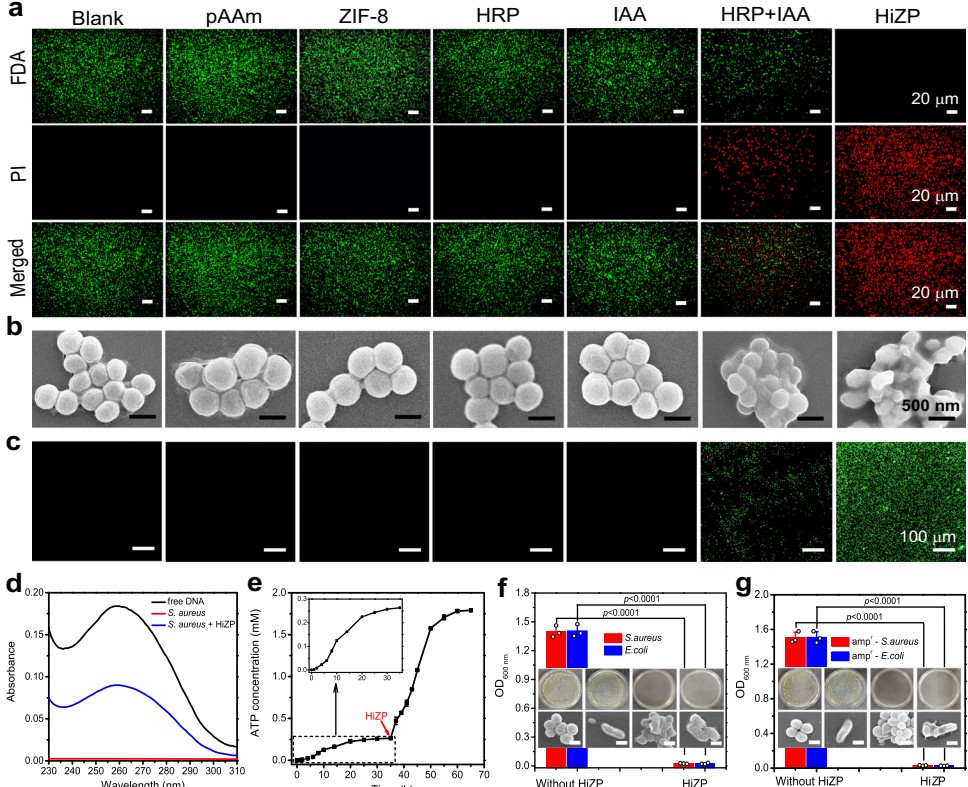

**Fig. 6 | Antibacterial mechanism of HiZP. a** Live/dead fluorescent staining images (scale bar = 20 μm) and (**b**) SEM images (scale bar = 500 nm) of *S. aureus* after different treatments. Green and red signals denote viable and dead bacteria, respectively. **c** Fluorescence images of DiBAC4(3) stained *S. aureus* after different treatments (scale bar = 100 μm). Green signal denotes bacterial membrane damage (depolarization). **d** Absorption spectra of free DNA and the supernatants of untreated *S. aureus* and *S. aureus* treated with HiZP. **e** Time-dependent changes in ATP concentrations before and after treating *S. aureus* with HiZP. The antibacterial behaviors of HiZP against (**f**) *S. aureus* and *E. coli* and (**g**) amp-resistant *S. aureus* and *E. coli*. Scale bars are 500 nm. The images in (**a**), (**b**), (**c**), (**f**), and (**g**) are representatives of three independent experiments with similar results. Data are presented as mean values ± standard deviation (*n* = 3 independent experiments). Statistical significance was analyzed via one-way ANOVA with Tukey's multiple comparisons test.

## Effects of HiZP on bacterial biofilms

Encouraged by its outstanding antibacterial ability, the potential of HiZP to eradicate biofilms was then investigated. Biofilms are a three-dimensional community of bacteria living within a secreted matrix of hydrated excreted extracellular polymeric substances comprised of lipids, polysaccharides, proteins, and extracellular DNA. Notably, it has been demonstrated that the formation of biofilms is often related to a broad range of persistent infections and drug resistance[43,44]. In this work, *S. aureus* was employed as a model to construct biofilms, and the biofilm mass was quantified by a standard crystal violet staining assay. Figure 7a shows that compared with untreated biofilm (blank), HiZP treatment caused a significant decrease (> 85%) in biofilm mass, a reduction that was also much lower than that generated by the free HRP/IAA system (41.6%). This result shows the excellent capability of HiZP to destroy biofilms, which was further demonstrated by the dose-dependent reduction in biofilm mass with increasing HiZP concentration (Fig. 7b). In contrast, almost no changes in biofilm mass were observed in the presence of HRP and IAA alone, even at high concentrations, indicating their negligible effects to disperse biofilms. Similarly, it was found that HiZP also inhibited the formation of *S. aureus* biofilms (Supplementary Fig. 21). Furthermore, by performing a live/dead fluorescent staining assay (Fig. 7c), we found that upon dispersion of the biofilm, the embedded *S. aureus* was effectively killed by HiZP. Taken together, HiZP effectively destroyed the established biofilm, inactivated the bacteria embedded in the biofilm and inhibited the formation of new biofilms. All of these effects originate from the oxidative damage to the biofilm components and bacterial membranes caused by ROS.

## Wound disinfection

The in vivo antibacterial activity of HiZP was examined by using a mouse model of *S. aureus*-infected wounds. First, wounds with diameters of approximately 1 cm were created on the backs of male Kunming mice (6-8 weeks) and then injected with $1 \times 10^6$ *S. aureus* to establish the infected wound model. From Fig. 8a, we can see that similar to uninfected wounds (blank), no erythema or edema appeared in the wounds treated with HiZP, and scabs formed within two days. This suggests that HiZP remains inactive in uninfected healthy wounds and has negligible adverse effects, which is consistent with the excellent biocompatibility of HiZP, as reflected by the high viability (more than 85%) of NIH/3T3 cells after treating with HiZP for 24 h (Supplementary Fig. 22) and in the co-culture system of NIH/3T3 with *S. aureus* and HiZP (Supplementary Fig. 23). However, four days after *S. aureus* injection, obvious erythema and edema appeared in the infected wounds that did not receive treatment, and the wound size increased ~2-fold (Supplementary Fig. 24). Treatment with the free HRP/IAA system showed less serious wounds, but a certain amount of erythema and edema remained in the infected wound. In contrast, after treatment with HiZP, the infected wound displayed no erythema or edema, and wound closure was observed after seven days of therapy, revealing the superior treatment efficacy of HiZP. To quantitatively assess the antibacterial effects of HiZP, the wound tissues in each group were collected after seven days of therapy to determine the number of surviving bacteria. The results from Fig. 8b and c show that HiZP produced the most effective antibacterial therapy in the wounds, and was much superior to the free HRP/IAA system.

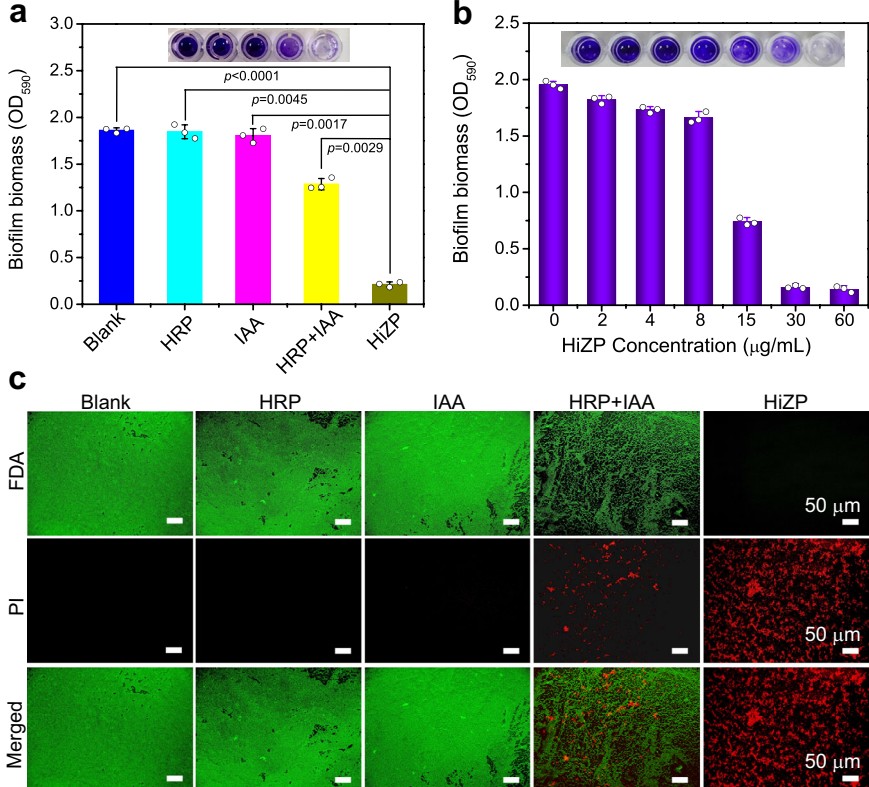

**Fig. 7 | Eradication of bacterial biofilms by HiZP. a** $OD_{590}$ values to evaluate the efficacy of eliminating *S. aureus* biofilms after treatment with HRP, IAA, mixture of HRP and IAA, and HiZP. **b** $OD_{590}$ values to evaluate the efficacy of eliminating *S. aureus* biofilms after treatment with different concentrations of HiZP. **c** Live/dead fluorescent staining images of the residual biofilms after different treatments. Scale bars are 50 μm. The images are representatives of three independent experiments with similar results. Data are presented as mean values ± SD ($n = 3$ independent experiments). Statistical significance was analyzed via one-way ANOVA with Tukey's multiple comparisons test.

The wound healing process was investigated using histological analysis (Fig. 8d), including hematoxylin and eosin (H&E) staining and Masson's trichrome staining. The H&E staining results showed that compared with uninfected wounds (blank), numerous inflammatory cells and a fragmented epidermis were present in the *S. aureus*-infected wounds. However, after treatment with HiZP, the infected wound displayed an intact epidermal layer and extremely low levels of inflammatory cells (on the same level as the uninfected wound), indicating that the infected wound had completely healed. Notably, treatment with the free HRP/IAA system also led to infected wound healing, but its efficacy was significantly inferior to that of HiZP. Since the formation of collagen is an important indicator of tissue regeneration[45], the therapeutic efficacy of HiZP was further evaluated by performing Masson's trichrome staining, in which collagen was stained blue. From Fig. 8d, we can see that most of the collagen was lost in the infected wounds, while the deposition of newly generated collagen was found in the infected wounds after treatment with HiZP. Importantly, the deposition density and organization of the regenerated collagen in the infected wound treated with HiZP were the same as those in the uninfected wound, indicating that the infected wound was completely healed.

In addition, histological analysis of major organs (heart, liver, spleen, lungs and kidneys) was performed to evaluate the long-term toxicity of HiZP in vivo. From the H&E staining results in Supplementary Fig. 25, we can see that compared with normal healthy mice (control), no obvious histological abnormalities or inflammatory lesions were found in the mice treated with HiZP. This suggest that HiZP has minimal toxic side effects in mice, a result that is supported by the small fluctuation in body weight in the mice treated with HiZP (Supplementary Fig. 26). Therefore, HiZP has good biocompatibility and can be used as a safe therapeutic material for wound healing.

In summary, an ATP-activated HRP/IAA prodrug system (HiZP) was fabricated for the on-demand treatment of bacterial infections. The synergistic actions of ZIF-8 and pAAm microspheres enabled HiZP to simultaneously transport HRP and IAA within a single carrier while avoiding the premature activation of IAA. Moreover, as an integral prodrug system, the superiority of HiZP was highlighted by its greatly enhanced ROS yields and subsequent antibacterial activity compared to the free HRP/IAA system. In contrast to conventional ROS systems[46–50], HiZP does not require toxic $H_2O_2$ to produce ROS and remains inactive in heathy tissues. Meanwhile, IAA is highly tolerant to mammalian peroxidases, which cannot oxidize IAA to generate ROS as HRP does. Therefore, by using HiZP as a prodrug system, an on-demand approach with negligible side effects was established for the effective treatment of bacterial infection. Importantly, after initiating the ROS generated by the ATP secreted by bacteria, the activation of HiZP can be accelerated by the leakage of intracellular ATP from dead bacteria, further improving the antibacterial efficacy. From the in vivo antibacterial results, the applicability of HiZP was demonstrated for wound disinfection. Accordingly, our presented prodrug strategy not only provides a robust way to solve the contradiction of simultaneous transport and premature activation of the traditional HRP/IAA prodrug system but also offers an opportunity for utilizing ATP as a metabolic trigger to develop on-demand antibacterial agents.

## Methods
### Synthesis of ZIF-8, FAM@ZIF-8 and IAA@ZIF-8
Briefly, 2 mL of aqueous solution containing 2-MeIM (500 mM) was stirred gently at room temperature, followed by the addition of 2 mL of

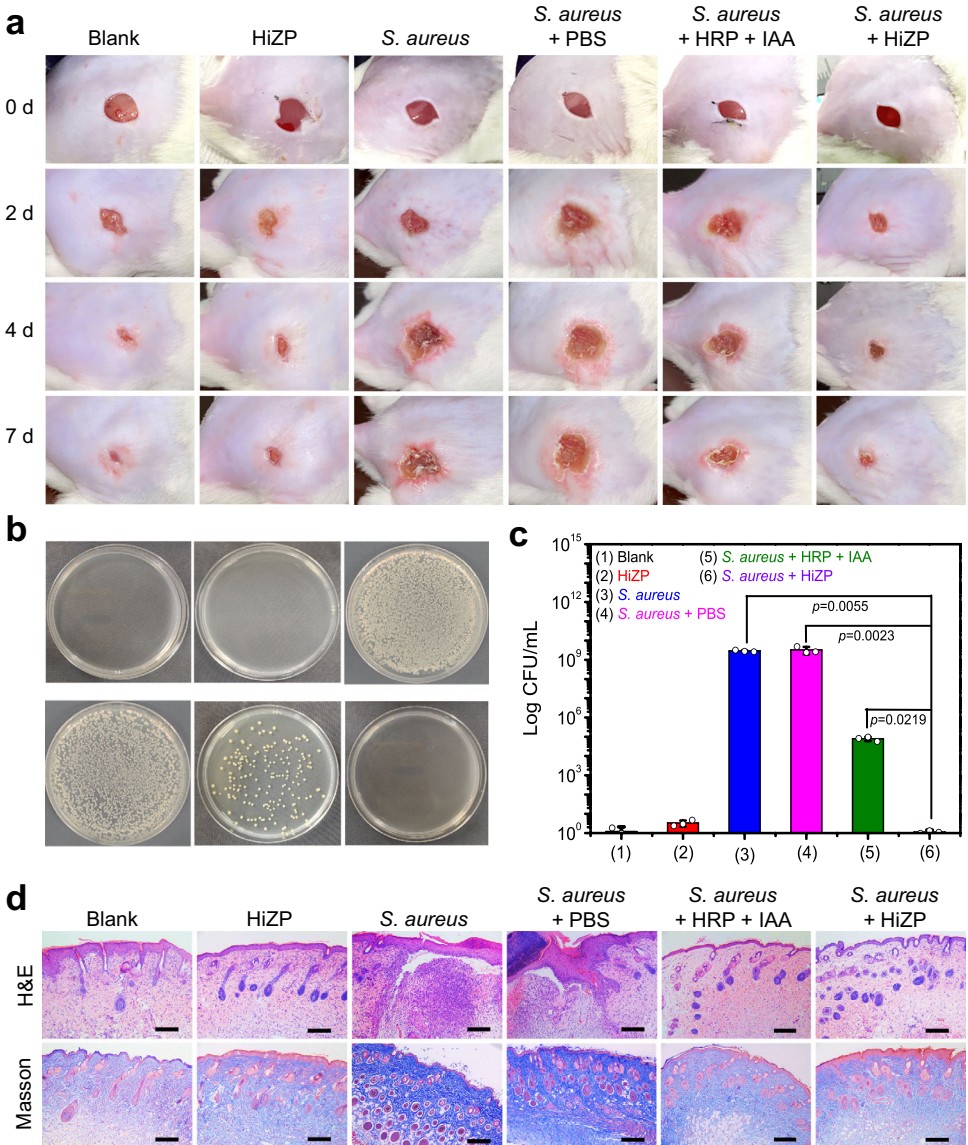

**Fig. 8 | Wound disinfection capability of HiZP on mouse model. a** Photographs of uninfected wounds and *S. aureus*-infected wounds treated with PBS, the free HRP/IAA system and HiZP. **b** Images of *S. aureus* colonies separated from the wound tissue plated on LB agar plates. **c** Number of surviving bacteria in the wound tissue of each sample. **d** Histological photomicrographs of skin tissue sections after H&E staining and Masson's trichrome staining. Scale bars are 100 μm. The images are representatives of three independent experiments with similar results. Dada are presented as mean values ± SD (*n* = 3 independent experiments). Statistical significance was analyzed via one-way ANOVA with Tukey's multiple comparisons test.

Zn(NO$_3$)$_2$·6H$_2$O aqueous solution (125 mM). The mixture stirred overnight. Then, the white product (ZIF-8) was collected by centrifugation and washed with ultrapure water three times. Finally, the as-synthesized ZIF-8 was dispersed in 1 mL of ultrapure water for further use.

The same synthetic conditions and procedures were used to prepare FAM@ZIF-8 and IAA@ZIF-8, but 2 mL of pure 2-MeIM solution was replaced by 2 mL of a mixture of 2-MeIM and 2 mg of FAM to produce FAM@ZIF-8 or replaced by 2 mL of a mixture of 2-MeIM and 30 mg of IAA to produce IAA@ZIF-8.

**Fabrication of HiZP**
The HiZP was prepared by using a T-type polydimethylsiloxane (PDMS) chip with a width of 1 mm and a depth of 0.50 mm. Typically, 1 mL of pregel aqueous solution was contained 15% (v/v) AAm, 1% (wt%) BIS, 1% (wt%) 2-hydroxy-4'-(2-hydroxyethoxy)-2-methylpropiophenone, 3 mg/mL HRP, and 1.5 mg/mL IAA@ZIF-8. The continuous (oil) phase was a paraffin liquid containing 3% (wt%) Span 80 as a surfactant to stabilize

the generated emulsion droplets. The flow speeds were 2 μL/min for the pregel and 20 μL/min for the oil stream. A UV filter (OmniCure S1500) was used to provide the desired excitation for gelation of the pAAm microspheres. Isopropanol was used to break the emulsion, and then ultrapure water was used to wash the microspheres. The final products were dispersed in 2 mL of ultrapure water and stored at −4 °C for further use.

**Protein measurements**
A classical BCA colorimetric assay was used to determine the protein (i.e. HRP) in HiZP. The BCA colorimetric assay was performed with a commercial kit (Thermo Scientific, 23227). The BCA working solution was prepared by mixing 50-parts BCA reagent 1 (that is comprised of sodium carbonate, sodium bicarbonate, BCA, and sodium tartrate in 0.1 M sodium hydroxide) and 1-part BCA reagent 2 (that is comprised of 4% cupric sulfate). Typically, HiZP (20 μg/mL) was suspended in 50 μL of PBS buffer (10 mM, pH 7.4), followed by the addition of 500 μL of BCA working solution. Then, the

mixture was reacted for 30 min at 37 °C. After cooling down room temperature, the absorbance spectra of the BCA reaction solutions were measured using a UV-Vis spectrophotometer (Hitachi, U-3900H, Japan). The same reaction conditions and procedures were used to prepare a standard curve for determining the concentration of HRP in HiZP, but HiZP was replaced by free HRP with different concentration (from 0 to 30 µg/mL).

## ROS measurements

A DCFH fluorescence assay was used to determine the generation of ROS. Typically, 15 µg/mL HiZP was first mixed with 100 µL of DCFH (40 µM). Then, different concentrations of ATP were added to the mixture. After reacting for 30 min at room temperature, the fluorescence spectra of the DCFH solutions were measured with an excitation wavelength of 488 nm (Hitachi, F-7000, Japan). Notably, ATP was not required to determine the amount of ROS generated from the free HRP/IAA system, and in this experiment, the concentrations of HRP and IAA were 10 µg/mL and 3 mg/mL, respectively.

## Antibacterial experiments

*S. aureus* was recovered from frozen glycerol stocks by inoculation in LB medium at 37 °C overnight under rotation at 170 rpm. The bacterial concentrations were quantified by measuring the optical density of the medium at 600 nm ($OD_{600\,nm}$). The bacteria were diluted with sterile 0.01 M PBS to $OD_{600\,nm} = 0.1$ (equivalent to $10^8$ CFU/mL) for further use.

For a typical antibacterial test, six groups of *S. aureus* suspensions ($OD_{600\,nm} = 0.1$) were first allowed to grow to $OD_{600\,nm} = 1.0$ and then incubated with (1) PBS, (2) pAAm, (3) ZIF-8, (4) HRP, (5) IAA, (6) HRP + IAA, or (7) HiZP. Incubation was carried out at 37 °C in a rotary shaker with rotation at 170 rpm. The concentrations of pAAm, ZIF-8, HRP, IAA, and HiZP were all 50 µg/mL. After incubation for 30 h, 50 µL of each bacterial suspension was mixed with 50 µL of sterile medium, and the bacterial numbers were counted. For the plate-counting method, the treated bacterial suspensions (100 µL) were spread on agar culture plates and incubated at 37 °C for 24 h. After incubation, the number of colonies was counted. All experiments were repeated three times. The bacterial survival rate was calculated as follows: survival rate (%) = Nt/Nc × 100%, where Nt is the number of colonies formed in the experimental group and Nc is the number of colonies formed in the blank group (PBS alone).

## Live/dead fluorescent staining

FDA and PI were employed for live/dead fluorescent staining. FDA was dissolved in acetone and stored at -20 °C, while PI was dissolved in PBS (pH = 7.4) and stored at 4 °C. Typically, *S. aureus* suspensions ($10^8$ CFU/mL) were treated separately with 50 µg/mL of one of the following: (1) PBS, (2) pAAm, (3) ZIF-8, (4) HRP, (5) IAA, (6) HRP + IAA, or (7) HiZP, followed by incubation at 37 °C for 24 h. The bacteria were collected by centrifugation and washed with 0.9% NaCl three times. Then, the bacteria were stained with 1.5 µL of 1% green-fluorescent nucleic acid stain (FDA) and 1.5 µL of 1% red-fluorescent nucleic acid stain (PI) for 15 min. Bacterial samples were imaged using a fluorescence microscope (Olympus, IX53, Japan).

## SEM measurements

Bacterial morphology was characterized by SEM imaging. Typically, after washing with PBS (10 mM, pH = 7.4), the bacteria were fixed with 2% glutaraldehyde for 3 h at 4 °C. Then, the fixed samples were successively dehydrated with 30, 50, 60, 70, 80, and 100% (v/v) ethanol for 10 min at each concentration. Finally, the samples were freeze-dried for SEM imaging with a scanning electron microscope (Hitachi, S-3400, Japan).

## Bacterial biofilm dispersion and inhibition

For bacterial biofilm dispersion, *S. aureus* was first seeded in a conical flask with LB medium and incubated with shaking at 37 °C. When the $OD_{600}$ value reached 1.0, the bacteria were collected and rinsed twice with sterile PBS (10 mM, pH = 7.4) to remove the culture medium. Then, the bacteria were resuspended in 0.9% physiological saline to an $OD_{600}$ value of 0.1 (equivalent to $10^8$ CFU/mL). After that, 100 µL of bacterial suspension was mixed with 100 µL of LB medium in a 96-well plate and further incubated at 37 °C without shaking to form biofilms. The culture medium in each well was replaced every 12 h. After 48 h, newly produced biofilms were obtained at the bottom of the wells. After removing the unbound bacteria and culture medium by rinsing with PBS, the obtained biofilms were then treated by adding 200 µL of LB medium with (1) PBS, (2) HRP, (3) IAA, (4) HRP + IAA, or (5) HiZP. The final concentrations of HRP, IAA, and HiZP were all 50 µg/mL in each well. After 48 h of incubation at 37 °C, the residual biofilms in the wells were rinsed with PBS, followed by separately staining with (1) live/dead fluorescence dyes (i.e. FDA/PI) for the determination of bacterial viability in the biofilms, and (2) crystal violet for biomass quantitation of the biofilms.

Different from the bacterial biofilm dispersion, the inhibition of bacterial biofilm formation was accomplished by treating *S. aureus* with (1) PBS, (2) HRP, (3) IAA, (4) HRP + IAA, or (5) HiZP before the formation of biofilms. Specifically, 100 µL of *S. aureus* suspension ($10^8$ CFU/mL) was placed in a 96-well plate and treated by adding 100 µL of LB medium with (1) PBS, (2) HRP, (3) IAA, (4) HRP + IAA, or (5) HiZP. The final concentrations of HRP, IAA, and HiZP were all 50 µg/mL in each well. Then, the 96-well plate was incubated at 37 °C without shaking. The culture medium in each well was replaced every 12 h. After 48 h of incubation, the biofilms in the wells were rinsed with PBS, followed by separately staining with (1) live/dead fluorescence dyes (i.e. FDA/PI) for the determination of bacterial viability in the biofilms, and (2) crystal violet for biomass quantitation of the biofilms.

To determine the bacterial viability in the biofilms after different treatment, live/dead fluorescent staining assay was performed. Typically, after rinsing the obtained biofilms with PBS, an aliquot of PBS (200 µL) with 1% FDA and 1% PI was added to each well for staining the biofilms. The staining assay was carried out at room temperature in the dark. After 15 min of staining, the PBS medium in the wells was removed, and the stained biofilms were imaged using a fluorescence microscope (Olympus, IX53, Japan).

For biomass quantitation of biofilms, crystal violet staining was performed as follows. First, 300 µL of 1% crystal violet was added to the wells to stain the biofilms. After 30 min of staining, the wells were rinsed with ultrapure water and thoroughly dried. Then, 500 µL of 95% ethanol was added to each well for 1 h of reaction at room temperature with shaking. Finally, the crystal violet solution was diluted, and the optical density at 590 nm ($OD_{590}$) was measured using a microplate reader (Molecular Devices, SpectraMax M2, USA).

## Measurement of bacterial membrane potential

DiBAC4(3) was employed as a fluorescence indicator for the measurement of bacterial membrane potential. DiBAC4(3) was dissolved in anhydrous dimethyl sulfoxide (DMSO) and stored at -20 °C in the dark. Typically, 100 µL of *S. aureus* suspensions ($10^8$ CFU/mL) was first placed in a 96-well plate and separately mixed with 100 µL of LB medium that contains (1) PBS, (2) pAAm, (3) ZIF-8, (4) HRP, (5) IAA, (6) HRP + IAA, or (7) HiZP. The concentrations of pAAm, ZIF-8, HRP, IAA, and HiZP were all 50 µg/mL in each well. After 48 h of incubation at 37 °C, the bacteria were rinsed with PBS to remove culture medium. Then, the bacteria were stained with 100 µL of 20 µM DiBAC4(3) in the dark, and the staining assay was lasted for 60 min at 37 °C. Finally, the stained bacterial samples were imaged using a confocal laser scanning microscope (Leica, TCS SP8, Germany). The emission spectra of the strained bacterial samples were recorded using a fluorescence

spectrophotometer (Hitachi, F-7000, Japan) at an excitation wavelength of 493 nm.

## Cytotoxicity test

The cytotoxicity of HiZP was tested by using a 3-(4,5-dimethylthiazol-2-yl)-2,5-diphenyltetrazolium bromide (MTT) assay. Mouse embryonic fibroblast (NIH/3T3) cells were employed as the model cells and were incubated in Dulbecco's modified eagle medium (DMEM) containing 10% fetal bovine serum (FBS), 100 U/mL penicillin and 100 µg/mL streptomycin. Typically, the NIH/3T3 cells were firstly seeded in a 96-well plate with a density of 5000 cells per well and incubated in a 5% $CO_2$ atmosphere for 24 h. After that, the cell medium was replaced by fresh cell medium containing different amounts of HiZP (from 0 to 120 µg/mL), and then the cells were incubated for another 24 h under identical conditions. Then, the cell medium in each well was removed, and followed by adding 10 µL of MTT (5 mg/mL) to each well for reacting another 4 h at 37 °C. Finally, the MTT solution was replaced with 100 µL of DMSO to dissolve the formazan crystals. After reacting for 10 min under gentle shaking, the absorbance at 570 nm of the plate wells were measured by a microplate reader for calculating cell viability. The cell viability was expressed as the percentage of viable cells after treatment with HiZP compared with the percentage of viable cells among untreated cells.

## Mouse wound model

All animal experimental procedures were in accordance with the guidelines of the National Institutes of Health for the Care and Use of Laboratory Animals and were approved by the Ethics Committee of College of Life Science at Jiangxi Normal University (No. 20211023001). Male Kunming mice (8 weeks old, 30-40 g) were bought from Laboratory Animal Science and Technology Center at Jiangxi University of Chinese Medicine and were raised in a pathogen-free room with a temperature of $25 \pm 3$ °C, a relative humidity of 60–70%, and a 12 h light/dark cycle. The mice were divided into six groups (three mice in each group): (1) PBS (blank), (2) HiZP, (3) *S. aureus*, (4) *S. aureus* + PBS, (5) *S. aureus* + HRP + IAA, and (6) *S. aureus* + HiZP. Wounds were generated on the backs of the mice. The infected wound model mice (Group 3 through Group 6) were injected with a $10^8$ CFU/mL *S. aureus* suspension. After 24 h of infection, 50 µg/mL HiZP or a mixture of HRP (10 µg/mL) and IAA (3 mg/mL) was placed on the wound areas in Group 6 and Group 5 via an alginate hydrogel patch (Supplementary Fig. 27). PBS alone was added to the wound of the mice in Group 4. The wounds were observed and photographed, and hydrogel patches were changed every 24 h. After treatment for 7 days, the wound tissues were harvested to quantify the number of bacteria for hematoxylin and eosin (H&E) staining and Masson's trichrome staining. Finally, the mice were dissected, and the main organs (heart, liver, spleen, lungs, and kidneys) were collected for H&E staining to analyze biocompatibility.

## Statistical analysis

All experiments were repeated three times independently, and the data were presented as the mean values ± standard deviations (SD). GraphPad prism 8, Origin 8.0 and Image J version 1.53c were used for data processing and statistical analysis. Statistical significance was analyzed via one-way ANOVA with Tukey's multiple comparisons test.

## Reporting summary

Further information on research design is available in the Nature Research Reporting Summary linked to this article.

## Data availability

The data generated in this study are provided in the Supplementary Information and Source Data files provided with this paper. Data is also available from the corresponding author upon request. Source data are provided with this paper.

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

## Acknowledgements

This work was supported by the National Natural Science Foundation of China (21765010, 22064011) and the Natural Science Foundation of Jiangxi Province (20202ACB205003). We also appreciate the assistances received from Prof. Liangguo Xu in cytotoxicity test, Prof. Zhigang Gong in animal experiment, and Dr. Hao Cui in the analysis of histopathology data.

## Author contributions

Y.W. and H. T. conceived and designed the research. Y.W., H.C. and X.C. performed the experiments. Y.W., H.C., X.C., C.H.C, and H.T. analyzed the data. H.Y. contributed to antibacterial experiments. C.H.C contributed to revising the manuscript. H.T. supervised the research and wrote and revised the manuscript. All authors discussed the results and commented on the manuscript.

## Competing interests

The authors declare no competing interests.
