## [Peer Review File · Nature Communications]

Adenosine triphosphate-activated prodrug system for on-demand bacterial inactivation and wound disinfectionREVIEWER COMMENTS

Reviewer #1 (Remarks to the Author):

The manuscript describes the development of a micrometric platform composed by a zeolitic framework capable to load the prodrug indol-acetic acid and HRP encapsulated into polyacrylamide hydrogels. This system shows stimuli-responsive behavior releasing the housed prodrug in response to ATP, which is common in bacterial infection. IAA release is transformed into ROS by HRP destroying the bacteria. The novelty of the system is modest because there are different nanoplatfroms which have employed the pair HRP/IAA in different applications, as the authors indicated in the introduction, but the excellent stimuli-responsive behavior achieved by this microplatform in combination with the nice antibacterial response enhance their impact.

The work is clearly presented, the results confirm the antibacteria properties of this system both in in vitro assays and in animal models. The characterization of the system has been carefully performed. In my opinion, this work present enough novelty and quality to be published in this journal without modifications.

Reviewer #2 (Remarks to the Author):

This manuscript presents an adenosine triphosphate (ATP)-activated prodrug system HRP&IAA@ZIF-8@pAAm and demonstrates its application for the treatment of bacterial infection. The prodrug IAA was encapsulated in the ZIF-8 to prevent the premature activation, and the stability and catalytic efficiency of HRP can be enhanced by the pAAm microspheres. These two points make this work interesting. I would recommend publication in Nature Communication. Some comments are listed below.

1. Figure 1c, the red and blue curves should not be denoted as FAM@ZIF-8 and FAM@ZIF-8 + ATP, respectively, because the emission spectra were measured using the supernatants.
2. Page 6, "To confirm this speculation" is not clear, please clearly state the speculation.
3. Figure 2c, the Abs. for the HiZP at 560 nm was higher than that for the free HRP, why? Also please clearly describe the detailed experiments.
4. The encapsulation efficiency of HRP was determined to be 87.05%, which is much higher than conventional HRP hosts such as porous silica (~15%), why?
5. Please add a reference for "Such different release behaviors may arise from.....to prevent the leakage of HRP from HiZP."
6. Please add a reference for the "confinement effects" of the pAAm microspheres or other nanocarriers.
7. All abbreviations should be taken care of when mentioning for the first time in the text such as "IAA".

Reviewer #3 (Remarks to the Author):

The authors prepared the adenosine ATP-activated prodrug system with excellent bacterial inactivation. However, some changes are necessary to make it publishable.

The detailed comments are as follows:

1. In your work, the HiZP microspheres are prepared by droplet-based microfluidic technology under UV. But, numerous studies have shown that UV light may lead to the reduction of iron in horseradish peroxidase and the passivation of HRP. Authors only compare the catalytic activity HiZP and free HRP, and thus deducted that the HRP catalytic activity did not suffer significant alterations during the HiZP fabrication process, and that is not persuasive. Authors ignore the complexity of components in HiZP and other possibilities. For example, refer to SCIENCE ADVANCES,2020, DOI: 10.1126/sciadv.aay9751, ZIF-8 have antioxidant activity by ABTS free radical scavenging assays.

2. Porous morphology of hydrogels likely to cause ZIF-8 leakage. SEM images of the HiZP hydrogels should be provided to prove the confinement of hydrogels.

3. Why authors chose the ratio of HRP (3 mg/mL) and IAA@ZIF-8 (1.5mg/mL)? Is it the proportion selected after optimization?

Responses to the Reviewers' Comments

Reviewer #1 (Remarks to the Author):

The manuscript describes the development of a micrometric platform composed by a zeolitic framework capable to load the prodrug indol-acetic acid and HRP encapsulated into polyacrylamide hidrogels. This system shows stimuli-responsive behavior releasing the housed prodrug in response to ATP, which is common in bacterial infection. IAA release is transformed into ROS by HRP destroying the bacteria. The novelty of the system is modest because there are different nanoplatforms which have employed the pair HRP/IAA in different applications, as the authors indicated in the introduction, but the excellent stimuli-responsive behavior achieved by this microplatform in combination with the nice antibacterial response enhance their impact.

The work is clearly presented, the results confirm the antibacteria properties of this system both in in vitro assays and in animal models. The characterization of the system has been carefully performed. In my opinion, this work present enough novelty and quality to be published in this journal without modifications.

Reply: Many thanks for your positive comments.

Reviewer #2 (Remarks to the Author):

This manuscript presents an adenosine triphosphate (ATP)-activated prodrug system HRP&IAA@ZIF-8@pAAm and demonstrates its application for the treatment of bacterial infection. The prodrug IAA was encapsulated in the ZIF-8 to prevent the premature activation, and the stability and catalytic efficiency of HRP can be enhanced by the pAAm microspheres. These two points make this work interesting. I would recommend publication in Nature Communication. Some comments are listed below.

1. Figure 1c, the red and blue curves should not be denoted as FAM@ZIF-8 and FAM@ZIF-8 + ATP, respectively, because the emission spectra were measured using the supernatants.

Reply: Many thanks for your positive comments. According to your suggestion, we have changed the legends of FAM@ZIF-8 and FAM@ZIF-8 + ATP to supernatant of FAM@ZIF-8 and supernatant of (FAM@ZIF-8 + ATP), respectively. Please see new Figure 1c.

2. Page 6, "To confirm this speculation" is not clear, please clearly state the speculation.

Reply: In page 6, the speculation means the presence of IAA@ZIF-8 in HiZP. We have added the statement in the revised manuscript.

3. Figure 2c, the Abs. for the HiZP at 560 nm was higher than that for the free HRP, why? Also please clearly describe the detailed experiments.

Reply: The absorption spectra in Figure 2c showed the results of the qualitative analysis for identifying the presence of HRP in HiZP. Theoretically, identical absorption spectra of BCA reagents will be measured by the addition of free HRP and HiZP with same concentration. So, to clearly distinguish the absorption spectrum of HiZP from that of free HRP, different concentrations of free HRP and HiZP were used for BCA assay. In the BCA assay, the concentrations of free HRP and HiZP are 10 and 20 $\mu\text{g/mL}$, respectively. In addition, we have added the detailed experiments in Experimental section.

4. The encapsulation efficiency of HRP was determined to be 87.05%, which is much higher than conventional HRP hosts such as porous silica (~15%), why?

Reply: Compared with porous silica, the higher encapsulation efficiency of HRP in HiZP could be mainly attributed to the electrostatic adsorption of HRP on the surface of ZIF-8, which results in the formation of a large-sized composite to prevent the free diffusion and leakage of HRP in HiZP (*Adv. Funct. Mater.* 2018, 28, 1705137; *Enzyme Microb. Tech.* 2008, 42, 235; *Nano Lett.* 2018, 18, 1448). In addition, the cell-like biocompatibility and the in situ and rapid solidification of pAAm hydrogel microsphere may also contribute to the high encapsulation efficiency of HRP in HiZP.

5. Please add a reference for “Such different release behaviors may arise from.....to prevent the leakage of HRP from HiZP.”

Reply: According your suggestion, we have added a reference for “Such different release behaviors may arise from.....to prevent the leakage of HRP from HiZP.”, please see new Ref. 32-34.

6. Please add a reference for the “confinement effects” of the pAAm microspheres or other nanocarriers.

Reply: According your suggestion, we have added a reference for the “confinement effects” of the pAAm microspheres or other nanocarriers, please see new Ref.37-39.

7. All abbreviations should be taken care of when mentioning for the first time in the text such as “IAA”.

Reply: Thanks for the helpful suggestion. We have checked all abbreviations in the manuscript very carefully.

Reviewer #3 (Remarks to the Author):

The authors prepared the adenosine ATP-activated prodrug system with excellent bacterial inactivation. However, some changes are necessary to make it publishable.

The detailed comments are as follows:

1. In your work, the HiZP microspheres are prepared by droplet-based microfluidic technology under UV. But, numerous studies have shown that UV light may lead to the reduction of iron in horseradish peroxidase and the passivation of HRP. Authors only compare the catalytic activity HiZP and free HRP, and thus deducted that the HRP catalytic activity did not suffer significant alterations during the HiZP fabrication process, and that is not persuasive. Authors ignore the complexity of components in HiZP and other possibilities. For example, refer to SCIENCE ADVANCES,2020, DOI: 10.1126/sciadv.aay9751, ZIF-8 have antioxidant activity by ABTS free radical scavenging assays.

Reply: Many thanks for your valuable comments. It is true that the HiZP microspheres are prepared by droplet-based microfluidic technology under UV. However, the UV-initiated gelation of the pregel droplet of HiZP was occurred in a fluid PTFE tube with a flow speed of 2 $\mu\text{L}/\text{min}$ (Fig. R1a), but not in a static state under UV light. In this case, the exposure time of each droplet under UV light (with a lighting coverage of $\sim 13 \text{ mm}^2$) is about 3 s. In addition, we also tested the effects of UV light on HRP activity by using a UV lamp with a power of 55W to irradiate HRP (1 $\mu\text{g}/\text{mL}$) for different times. The results (Fig. R1b) show that HRP activity remains almost unchanged under UV light, even after exposure to UV light for 9 h. Similarly, almost unchanged HRP activity was also observed from HiZP and the mixture of HRP and ZIF-8. Therefore, in our study, the UV light may show a negligible influence on the HRP activity in HiZP.

Certainly, ZIF-8 has an antioxidant activity by ABTS free radical scavenging assays, as demonstrated in the previous report (*Sci. Adv.* 2020, 6, eaay9751). This is also verified by the dose-dependent decrease of $\text{ABTS}^{+\cdot}$ absorbance in the presence of ZIF-8 (Fig. R1c). Nevertheless, from Fig. R1c, we can see that the obvious decrease of $\text{ABTS}^{+\cdot}$ absorbance was only occurred when ZIF-8 concentration reached to 1 mg/mL . By contrast, almost no changes in the $\text{ABTS}^{+\cdot}$ absorbance were recorded when ZIF-8 concentration is less than 1 mg/mL , even if the reactions were lasted for 5 h. In our study, the concentration of ZIF-8 in each HiZP is about 0.18 $\mu\text{g}/\text{mL}$, which is much lower than the threshold concentration (1 mg/mL) of ZIF-8. In addition, in our study, the catalytic activity of HRP was determined by monitoring the absorbance of $\text{ABTS}^{+\cdot}$ within 3 min, which represents the initial rate of the reaction. This reaction time (3 min) is much shorter than the initiation time (60 min) for the ABTS free radical scavenging assay of 1 mg/mL ZIF-8 (Fig. R1c), which is consistent with previous report (*Sci. Adv.* 2020, 6, eaay9751). Accordingly, the presence of ZIF-8 may show very limited effects on the application of ABTS colorimetric assay for the evaluation of HRP activity in HiZP. Indeed, the ABTS colorimetric assay has been extremely used as a very common method to characterize the catalytic activity of the integrated HRP (or peroxidase mimetics) with ZIF-8 (such as *Anal. Chem.* 2016, 88,

5489; *Chem. Sci.*, 2019, 10, 7852; *ACS Sustainable Chem. Eng.* 2019, 7, 17, 14611; *ACS Catal.* 2020, 10, 5949).

Furthermore, to obtain a more direct measurement of the structural changes of HRP, we measured the fluorescent emission of tryptophan buried in the interior of the enzyme. Because the emission of tryptophan is highly sensitive to the conformational change (*J. Am. Chem. Soc.* 2017, 139, 6530; *Nano Lett.* 2020, 20, 6630; *Chem. Eng. J.* 2021, 425, 131482), the structural change of HRP can be identified by a fluorescence spectroscopy. From Fig. R1d, we can see that native HRP shows a typical tryptophan emission maximum at 331 nm, while a significant red-shift was observed from the treated HRP with 8 M urea, a well-known unfolding agent. By contrast, compared with native HRP, the tryptophan emissions of HiZP and the mixture of HRP and ZIF-8 display almost no changes. This indicates that ZIF-8 has no influence on the conformation of HRP and the HRP in HiZP maintains its native structure. Accordingly, the HRP in HiZP can exhibit its original catalytic activity. We have added the fluorescence measurement in the revised manuscript, please see new Fig. S11.

Figure R1. (a) Schematic illustration of the droplet-based microfluidic technology for the fabrication of HiZP. (b) Absorbance of ABTS^{++} obtained from the $\text{ABTS-H}_2\text{O}_2$ colorimetric assays in the presence of UV-treated HRP, UV-treated mixture of HRP and ZIF-8, and UV-treated HiZP. (c) Effects of ZIF-8 on the absorbance of ABTS^{++} obtained from the $\text{ABTS-H}_2\text{O}_2$ colorimetric assays and corresponding color changes of the reaction solutions. (d) Fluorescent emission spectra of HRP, HiZP, treated HRP with 8 M urea, and the mixture of HRP and ZIF-8.

2. Porous morphology of hydrogels likely to cause ZIF-8 leakage. SEM images of the HiZP hydrogels should be provided to prove the confinement of hydrogels.

Reply: According to your suggestion, we measured the cross-sectional SEM image of HiZP to prove the confinement of IAA@ZIF-8 in pAAm hydrogel. From Fig. R2a, we can see that different from pure pAAm hydrogel with smooth pores, the pores of HiZP have a rough surface that was coated with many nanoparticles, implying the presence of IAA@ZIF-8 in pAAm hydrogel. To confirm this, the EDS mapping of Zn element was further measured. As shown in Fig. R2b, it was found that HiZP has a characteristic EDS mapping of Zn element, while no Zn element was observed in the pure pAAm. This reveals that the rough surface in HiZP is resulted from the deposition of IAA@ZIF-8 in pAAm hydrogel. In addition, by loading FAM@ZIF-8 in pAAm hydrogel, we found that HiZP can exhibit a characteristic green fluorescence of FAM, while no fluorescence was recorded from pure pAAm hydrogel (Fig. R2c). This fluorescence image is consistent with the results from EDS mapping and demonstrates the confinement of ZIF-8 in pAAm hydrogel. Nevertheless, after treating with ATP, the HiZP displays similar smooth pores as that of pure pAAm hydrogel, and no characteristic Zn element and FAM fluorescence were measured. These results further indicate that ZIF-8 is highly sensitive to ATP, which can destroy ZIF-8 framework to release the preloaded guest molecules (i.e., FAM or IAA). We have added the cross-sectional SEM image and the EDS mapping of Zn element of HiZP in the revised manuscript, please see new Fig. S9.

Figure R1. Cross-sectional SEM image (a) and EDS mapping of Zn element (b) of pure pAAm hydrogel, HiZP and treated HiZP with ATP. (c) Confocal laser scanning microscopy images of pure pAAm hydrogel, pAAm hydrogel loaded with FAM@ZIF-8 (denoted as FAM@ZIF-8@pAAm) and treated FAM@ZIF-8@pAAm with ATP.

3. Why authors chose the ratio of HRP (3 mg/mL) and IAA@ZIF-8 (1.5mg/mL)? Is it the proportion selected after optimization?

Reply: Yes. The ratio of HRP and IAA@ZIF-8 are selected proportion after optimization. Theoretically, more loading amounts of HRP and IAA@ZIF-8 in HiZP can produce a higher level of ROS to inactivate bacteria more efficiently. However, it is extremely difficult to be achieved for the fabrication of HiZP. This is because that the needle ($\Phi 0.31 \times 0.6$ mm) and PTFE tube ($\Phi 0.56 \times 1.06$ mm) that connect PDMS device are easily to be blocked if there is too much IAA@ZIF-8 nanoparticles in the pregel solution. Although less IAA@ZIF-8 can cause a uniform and stable flow of pregel solution in the needle and PTFE tube, the ROS yields of the resulting HiZP are usually insufficient to effectively inactivate bacteria. Similar to IAA@ZIF-8, the presence of less HRP in pregel solution could also result in insufficient supply of ROS to reduce the antibacterial efficacy. Therefore, optimized amounts and ratio of HRP and IAA@ZIF-8 were used to fabricate HiZP for achieving the best antibacterial efficacy.

REVIEWER COMMENTS

Reviewer #2 (Remarks to the Author):

I think that this revised version can be accepted.

Reviewer #3 (Remarks to the Author):

The paper is very much improved and I have no problem in recommending it for publication.

Reviewer #4 (Remarks to the Author):

In this manuscript authors tried to explain ATP mediated prodrug system to generate intracellular ROS and killing of bacterial pathogens. The manuscript has many flaws. The conclusion drawn is very straight forward without having enough experimental evidence. The experimental section is not clear and some of the results are not conclusive. I have following comments.

1. Experimental methods explained the ROS production in presence of ATP, however intracellular ROS production assay is not clear. The intracellular ROS production in bacterial cells upon treatment with HiZP in absence of ATP need to be mentioned properly.
2. Figure 2 was not collected properly. The baseline correction should be done with blank sample.
3. The intracellular ROS production in bacterial cell is not conclusive. The ROS production should be demonstrated by confocal or flow cytometry assay.
4. What is the surface charge of HiZP?
5. Is there any specific reason for production of ROS in bacterial cells only. Otherwise, same ROS will be generated in mammalian cells also in presence of HiZP.
6. The cytotoxic effect of HiZP to mammalian cells must be tested before animal assay. I am wondering how the in vivo animal experiments were conducted without in vitro cytotoxic data.
7. The target specific ROS generation should be tested in co-culture system.
8. It is claimed that ATP concentration is higher and increases with increasing concentration of HiZP. However, reverse should be there, as dead bacteria is metabolically inactive. Therefore, dead bacteria must have low ATP compared to the live cells.
9. How the fluorescence microscopy image of biofilm was recorded? The fluorescence microscopy images shown in the manuscript are not representative image of the biofilm. The images show planktonic cells only.
10. The concentration of HRP, IAA and HiZP is not consistent. Some places it is ug/ml and other places it is ul/ml.
11. The SEM image does not show any member damage upon treatment with HiZP. It shows agglomeration only. The bacterial cell membrane damage should be validated by alteration of cell membrane potential upon treatment with HiZP.
12. There is serious concern in figure 7C. How the blank and HiZP samples have bacterial colony. The Log CFU value of blank and HiZP samples must be zero.
13. The histopathology data do not show any necrosis of tissue sample upon bacterial infection. Moreover, no bacterial cell is visible in tissue samples upon infection with *S. aureus*.
14. Why polyacrylamide is used? Polyacrylamide is highly toxic.
15. The standard error values are very small in all results. In my experience I have never seen such small standard error values in biological systems. There is also large variation in individual experiments.
16. None of the data have p-value calculation.

Responses to Reviewers' Comments

Reviewer #2 (Remarks to the Author):

I think that this revised version can be accepted.

Reply: We appreciate reviewer's positive comment.

Reviewer #3 (Remarks to the Author):

The paper is very much improved and I have no problem in recommending it for publication.

Reply: Thanks for your encouraged comment.

Reviewer #4 (Remarks to the Author):

In this manuscript authors tried to explain ATP mediated prodrug system to generate intracellular ROS and killing of bacterial pathogens. The manuscript has many flows. The conclusion drawn is very straight forward without having enough experimental evidence. The experimental section is not clear and some of the results are not conclusive. I have following comments.

1. Experimental methods explained the ROS production in presence of ATP, however intracellular ROS production assay is not clear. The intracellular ROS production in bacterial cells upon treatment with HiZP in absence of ATP need to be mentioned properly.

Reply: Thanks for your valuable comments. In this work, the ROS production is resulted from the oxidation of IAA by HRP, which was occurred within HiZP microspheres, but not within bacterial cells. The oxidation reaction of IAA with HRP in the HiZP was initiated by ATP. Specifically, in the presence of ATP, the encapsulated ZIF-8 framework in HiZP was decomposed to release the preloaded IAA from IAA@ZIF-8 and react with the coconfined HRP, leading to the production of ROS. Since ATP is a typical secretion of bacteria, the presence of bacteria can also activate HiZP to produce ROS as that of free ATP. The produced ROS from the HiZP can destruct bacterial membrane due to its strong oxidative ability and consequently cause bacteria inactivation. Accordingly, HiZP was functionalized as an on-demand prodrug system for bacterial inactivation.

IAA represents a typical nontoxic prodrug for therapy, which can be oxidized by HRP to produce ROS with high toxicity. However, the therapeutic efficacy of the IAA/HRP system has often been restricted by

limited ROS generation due to the different accumulation and release behaviors of IAA and HRP, which usually causes distinct spatiotemporal distributions in target sites. Although this issue could be solved by the simultaneous transport of IAA and HRP in a single carrier, the premature activation of IAA by HRP easily occurs to cause side effects in healthy tissues. Therefore, the theme of this work is to develop a new IAA/HRP prodrug system to address the issue of the premature activation of IAA prodrug while achieving the simultaneous transportation of IAA and HRP in a single carrier. With this regard, this manuscript was mainly focused on the design, fabrication and characterization of HiZP from the point of material research.

In this manuscript, the HiZP was fabricated by the coencapsulation of IAA@ZIF-8 and HRP in pAAm microspheres. The pAAm microspheres were demonstrated to enhance the stability and catalytic efficiency of HRP while offering a confined space to encapsulate IAA@ZIF-8 and HRP. This leads to the simultaneous transportation of IAA and HRP in a single carrier. As a host of loading IAA, the ZIF-8 not only effectively entrap IAA from leaking, but also enables the preloaded IAA to be physically isolated from HRP to avoid the premature activation of IAA due to its size selectivity. Besides, ZIF-8 was demonstrated to possess a highly specific responsivity to ATP, which can destroy the framework structure of ZIF-8 to cause IAA leakage. Benefiting from the ATP-responsive characteristic of ZIF-8, the HiZP can be activated by ATP to generate ROS, which results from the ATP-triggered decomposition of the ZIF-8 framework and release of the preloaded IAA to react with the coconfined HRP in the pAAm microspheres (i.e. HiZP). Therefore, different from conventional IAA/HRP prodrug systems, HiZP is an integral prodrug system with stimuli-responsive behavior due to the synergistic actions of ZIF-8 and pAAm microspheres. On this basis (the successful fabrication of ATP-activated HiZP prodrug system), by considering the fact that bacteria generally secrete ATP during growth, the potential of HiZP as an on-demand prodrug system was explored for bacterial inactivation in this manuscript.

2. Figure 2 was not collected properly. The baseline correction should be done with blank sample.

Reply: Many thanks for your kind suggestion, we have corrected the baseline and a new Figure 2c has been presented in the revised manuscript.

3. The intracellular ROS production in bacterial cell is not conclusive. The ROS production should be demonstrated by confocal or flow cytometry assay.

Reply: As mentioned in Query No. 1, the production of ROS was resulted from the oxidation of IAA by HRP within HiZP microspheres, which was initiated by the addition of ATP. To confirm the production of ROS from HiZP, we have performed a classic fluorescence assay of 2',7'-dichlorofluorescein diacetate (DCFH) in this manuscript. DCFH is a well-known fluorescent probe for the sensitive detection of ROS, which itself is nonfluorescent but it can be oxidized by ROS to produce highly fluorescent 2',7'-dichlorofluorescein (DCF)

(*J. Photochem. Photobiol. B: Biol.* 2002, 67, 23; *Biochem. Biophys. Res. Commun.* 2010, 397, 603; *Adv. Mater. Technol.* 2017, 2, 1700033). From Fig. 3a, we can see that HiZP is not capable of oxidizing DCFH. However, after the addition of the mixture of ATP and HiZP, obvious fluorescence from DCF was measured from the solution originally containing DCFH, signifying the generation of ROS from the interaction between ATP and HiZP. Meanwhile, a time-dependent fluorescent response of HiZP to ATP was observed (Fig. S16). On this basis, by plotting the DCF fluorescence intensity against ATP concentration (Fig. 3c), we found that the DCF fluorescence gradually increased with increasing ATP concentration. Apparently, these results reveal the production of ROS from HiZP due to the triggering effect of ATP, which destroys the structure of ZIF-8 to release the preloaded IAA from IAA@ZIF-8 and thereby initiates the oxidation reaction of IAA with HRP to generate ROS. The dissociation of IAA@ZIF-8 has been verified by the vanishing characteristic EDS mapping of Zn element (Fig. S9) and XRD pattern (Fig. 2b) of ZIF-8 from HiZP.

4. What is the surface charge of HiZP?

Reply: The surface charge of HiZP is +2.76 mV, which is opposite to the negatively charged surface of *S. aureus* (-20.33 mV). Such opposite surface charges enable HiZP to bind *S. aureus* through electrostatic interactions and consequently inactivates bacterial more efficiently (*Adv. Mater.* 2011, 23, 4805; *Chem. Commun.* 2014, 50, 9298; *ACS Appl. Mater. Interfaces* 2017, 9, 9260; *J. Mater. Chem. B* 2021, 9, 125). We have added the data in the revised manuscript, please see p.12.

5. Is there any specific reason for production of ROS in bacterial cells only. Otherwise, same ROS will be generated in mammalian cells also in presence of HiZP.

Reply: In this manuscript, the ROS is produced from the oxidation of IAA by HRP, which was occurred in HiZP microspheres and initiated by ATP. Since ATP is a typical secretion of living bacteria, the presence of bacteria can also activate HiZP to generate ROS. Owing to the strong oxidative ability of ROS, the generated ROS can disrupt bacterial membrane to inactivate bacteria.

6. The cytotoxic effect of HiZP to mammalian cells must be tested before animal assay. I am wondering how the in vivo animal experiments were conducted without in vitro cytotoxic data.

Reply: We tested the cytotoxic effect of HiZP to mouse embryonic fibroblast (NIH/3T3) cells by using a standard MTT assay. The results (Fig. R1) show that HiZP has an excellent biocompatibility, and it does not exhibit obvious cytotoxicity to NIH/3T3 cells, even if the HiZP was presented at a high concentration (120 $\mu\text{g/mL}$). We have added the results in the revised manuscript, please see new Fig. S22.

Figure R1. Viability of the NIH/3T3 cells treated with different concentrations of HiZP.

7. The target specific ROS generation should be tested in co-culture system.

Reply: According to your suggestion, we tested the ROS generation from HiZP in a co-culture system by employing the classic DCFH fluorescence assay. The results (Fig. R2a) show that no DCF (oxidized product of DCFH by ROS) fluorescence was recorded from pure LB medium (without *S. aureus*), reflecting that no ROS was existed in pure LB medium. Similarly, the LB medium with HiZP also display no measurable DCF fluorescence, indicating that pure LB medium has no ability to trigger HiZP to generate ROS. However, after the introduction of *S. aureus*, a typical DCF fluorescence was measured from the LB medium with HiZP. This reveals that like free ATP, presence of *S. aureus* can trigger HiZP to generate ROS, which is arisen from the secretion of ATP from *S. aureus*. Since more ATP can be secreted from *S. aureus* upon the increase of incubation time, a time-dependent increase in the DCF fluorescence was observed (Fig. R2b). Therefore, it is feasible to activate HiZP to generate ROS in a co-culture system due to the secretion of ATP from *S. aureus*, which causes the decomposition of the ZIF-8 framework and release of the preloaded IAA to react with the coconfined HRP in HiZP.

Figure R2. (a) Emission spectra of DCFH after different treatments in LB medium. (b) Time-dependent DCF fluorescence in a co-culture system with *S. aureus*.

8. It is claimed that ATP concentration is higher and increases with increasing concentration of HiZP. However, reverse should be there, as dead bacteria is metabolically inactive. Therefore, dead bacteria must have low ATP compared to the live cells.

Reply: It is true that dead bacterium is metabolically inactive, and thereby it has a lower concentration of ATP secretion as compared to living bacterium. However, in this work, the ATP-triggered activation of HiZP was occurred before the bacteria were killed. At this point, the bacteria still remain activated and can secrete ATP to trigger the activation of HiZP. Accordingly, ROS was produced upon the activation of HiZP by the secreted-ATP. Owing to the strong oxidative capability of ROS, the generated ROS can destruct bacterial membrane, leading to bacterial inactivation and the leak of intracellular ATP from the killed bacteria. In this case, the leaked intracellular-ATP will accelerate the activation of HiZP to produce a large amount of ROS, and consequently cause more bacterial inactivation and leak more intracellular-ATP from the killed bacteria. Therefore, it is possible to observe increased ATP concentration with increasing the concentration of HiZP. Indeed, the increased ATP concentration is mainly resulted from the contribution of leaked intracellular-ATP from dead bacteria due to the destruction of bacterial membrane induced by ROS.

9. How the fluorescence microscopy image of biofilm was recorded? The fluorescence microscopy images shown in the manuscript are not representative image of the biofilm. The images show planktonic cells only.

Reply: The fluorescence microscopy image of the biofilm was recorded from the well-washed biofilm with PBS buffer (10 mM, pH 7.0) after staining with FDA and PI. The biofilm was immersed in PSB buffer when recording the fluorescence microscopy image. To avoid question, we have re-recorded the fluorescence microscopy image of biofilm deposition without PBS buffer, and the results have been added as new Fig. 6c and Fig. S21c in the revised manuscript.

10. The concentration of HRP, IAA and HiZP is not consistent. Some places it is ug/ml and other places it is ul/ml.

Reply: There is a typo in the concentrations of HRP, IAA and HiZP. The concentrations of HRP, IAA and HiZP are all $\mu\text{g/mL}$ in this manuscript. We have corrected the errors in the revised manuscript.

11. The SEM image does not show any member damage upon treatment with HiZP. It shows agglomeration only. The bacterial cell membrane damage should be validated by alteration of cell membrane potential upon treatment with HiZP.

Reply: According to your suggestion, we measured the cell membrane potential to further validate the bacterial cell membrane damage by employing DiBAC4(3) as a membrane potential indicator. The DiBAC4(3) is a slow-response potential-sensitive probe, and it can enter depolarized cells where it binds to

intracellular proteins or membrane and exhibits enhanced fluorescence. The fluorescence intensity of DiBAC4(3) is membrane potential-dependent (*J. Microbiol. Methods* 2001, 47, 233; *LWT-Food Sci. Technol.* 2019, 101, 100; *Food Chem.* 2021, 363, 130340; *Food Control* 2022, 131, 108435). By recording the fluorescence with confocal laser scanning microscopy, a strong green fluorescence can be observed from the treated bacteria with HiZP (Fig. R3a and R3b), reflecting the alteration of bacterial membrane potential as the bacterial membrane damage. Upon the increase of HiZP concentration, gradually increased fluorescence intensity of DiBAC4(3) was also measured (Fig. R3c). Although green fluorescence spots can also be recorded from the bacteria treated with free HRP/IAA system, its fluorescence intensity and density are much lower than that of treated bacteria with HiZP. This reveals that free HRP/IAA system has a weaker effect on bacterial membrane damage, which is consistent with its inferior antibacterial activity to HiZP. In addition, no fluorescence was measured from the untreated bacteria and the treated bacteria with individual pAAm, ZIF-8, IAA, and HRP, indicating their negligible toxicities to bacteria. We have added the result in the revised manuscript, please see new Fig. 5c and Fig. S19.

Figure R3. (a) Fluorescence images of DiBAC4(3) stained *S. aureus* after different treatments. Green signal denotes bacterial membrane damage (depolarization). (b) Fluorescence spectra of DiBAC4(3) measured from the stained *S. aureus* after different treatments. (c) Fluorescence spectra of DiBAC4(3) measured from the stained *S. aureus* after treating with different concentration of HiZP.

12. There is serious concern in figure 7C. How the blank and HiZP samples have bacterial colony. The Log

CFU value of blank and HiZP samples must be zero.

Reply: Certainly, the log CFU values of blank and HiZP samples should be zero. However, log (0) is invalid and not defined. The real logarithmic function log(x) is defined only for $x > 0$. In this work, the average x for blank and HiZP samples are 1.19 and 1.13, respectively. The values of log (1.19) and log (1.13) were calculated to be 0.07 and 0.05, respectively, which are extremely approached to zero. Since no less than 1 of bacterial colony was existed in real situation, the log CFU values of blank and HiZP samples are actually zero. Moreover, it should be pointed out that the scale of Y axis in original Fig.7c was started from log (0.5) with a log value of -0.3, but not from log (1) with a log value of 0. To avoid question, we have changed the start point of the scale of Y axis to log (1). Therefore, a new Fig.7c has been presented in the revised manuscript.

13. The histopathology data do not show any necrosis of tissue sample upon bacterial infection. Moreover, no bacterial cell is visible in tissue samples upon infection with *S. aureus*.

Reply: Thanks for your kind suggestion. We checked the histopathology data very carefully and recognized that the previous images are faulty, which is due to the inexact selection of the imaging region from the histological slides. Accordingly, new histological photomicrographs were taken from original histological slides in the revised manuscript to display the bacterial infection to skin tissues, please see new Fig.7d.

Normally, bacteria have a much smaller size as compared to tissues. For example, the diameter of *S. aureus* is ~ 550 nm, which is hardly to be observed by naked eyes, even by a common optical microscopy. Accordingly, to test the presence of bacteria in the infected wound, the most commonly used method is to excise the infected wound tissues for culturing and colony counting, which is also known as wound culture (*Biomaterials* 2019, 208, 21; *iScience* 2020, 23, 101281; *ACS Appl. Mater. Interfaces* 2021, 13, 40302; *Angew. Chem. Int. Ed.* 2021, 60, 3469; *Adv. Sci.* 2022, 9, 2104576). In addition, the complicated and tedious preparation procedure of histological slide, which involves five main stages: fixing, processing, embedding, sectioning and staining, could also lead to the loss of bacteria from the wound tissues. Therefore, in this work, the bacteria in the infected wound tissues were detected and quantified by a colony counting method, and the results were presented in Fig. 7b and 7c.

14. Why polyacrylamide is used? Polyacrylamide is highly toxic.

Reply: Previous studies have been demonstrated that polyacrylamide (pAAm) is non-toxic, while acrylamide monomer is a toxic substance that is capable of producing axonopathy by transection of neurons (*Rev. Environ. Health* 1991, 9, 215; *Water Sci. Technol.* 1998, 38, 207; *J. Hazard. Mater.* 2010, 175, 955). Indeed, pAAm product has long been the United States Environmental Protection Agency or the Food and Drug Administration approved for drinking water, juice clarification, fruits, vegetables and washing areas.

Generally, the toxicity of polyacrylamide is mainly originated from the residual toxicity of acrylamide. In this work, the pAAm hydrogel microsphere (i.e. HiZP) has been thoroughly washed with PBS buffer to remove unreacted acrylamide monomer, which is a highly water-soluble vinyl monomer.

In addition, we tested the cytotoxicity of the pAAm hydrogel microsphere by using a standard MTT assay. The results (Fig. R4) show that more than 95% of viability can be observed from NIH/3T3 cells after treating with pAAm hydrogel microsphere, even if the concentration of pAAm hydrogel microsphere was reached to 300 $\mu\text{g}/\text{mL}$. This indicates that the pAAm hydrogel microsphere possesses an excellent biocompatibility. Therefore, it is feasible to use pAAm as a host of loading IAA@ZIF-8 and HRP to fabricate HiZP prodrug system.

Figure R4. Viability of the NIH/3T3 cells treated with different concentrations of pAAm.

15. The standard error values are very small in all results. In my experience I have never seen such small standard error values in biological systems. There is also large variation in individual experiments.

Reply: In this manuscript, all experiments were conducted in triplicate for each sample, and the data were expressed by the mean and standard deviation. Error bars represent the standard deviation from the mean ($n=3$). We have added a section of Statistical Analysis in the revised manuscript. Indeed, similar to this work, small standard error values have also been observed in previously reported antibacterial systems based on various nanomaterials (*ACS Nano* 2015, 9, 2390; *Small* 2016, 12, 6200; *ACS Appl. Mater. Interfaces* 2017, 9, 16834; *Angew. Chem. Int. Ed.* 2021, 60, 3469; *Nat. Commun.* 2021, 12, 745).

16. None of the data have p-value calculation.

Reply: We calculated and indicated the p-values in the revised manuscript.

REVIEWER COMMENTS

Reviewer #4 (Remarks to the Author):

I have gone through the authors responses and noticed that authors did not respond all the comments satisfactorily. Authors are suggested to revise the manuscript. Below are my specific comments.

Q1. It was suggested to test the target specific ROS generation in co-culture system. However, authors did not respond this comments appropriately, might be they did not understand the comment. So I repeat the comment and suggested the authors to test the target specific ROS generation and cell viability assay in co-culture system of bacteria and fibroblast cells.

Q2. Authors responded that fluorescence microscopy image was taken without washing the biofilm deposition in PBS buffer. However, it is not clear how they prepared the biofilm. Whether the biofilm was formed on the multiwell plate and visualized under the fluorescence microscopy following staining with dyes or biofilm was formed on the glass slides and biofilm was imaged under the fluorescence microscopy following staining with dyes. Authors are suggested to clearly rewrite the experimental section.

Q3. The membrane depolarization assay, measured by membrane potential indicator dye DiBAC4(3), is not mentioned in the experimental section.

Responses to Reviewers' Comments

Reviewer #4 (Remarks to the Author):

I have gone through the authors responses and noticed that authors did not respond all the comments satisfactorily. Authors are suggested to revise the manuscript. Below are my specific comments.

Q1. It was suggested to test the target specific ROS generation in co-culture system. However, authors did not respond this comments appropriately, might be they did not understand the comment. So I repeat the comment and suggested the authors to test the target specific ROS generation and cell viability assay in co-culture system of bacteria and fibroblast cells.

Reply: Thanks for your kind explanation and valuable comments. Certainly, we had a misunderstanding to the previous comment on the co-culture system. Accordingly, some new experiments were performed to test the target specific ROS generation and cell viability assay in a co-culture system of bacteria (i.e. *S. aureus*) and fibroblast cells (i.e. NIH/3T3 cells), and the results were presented in Fig. R1. This co-culture system was constructed by mixing NIH/3T3 cells with *S. aureus* in a 96-well plate, which was incubated in a 5% CO₂ atmosphere, as required by pure NIH/3T3 cells. The initial densities of NIH/3T3 cell and *S. aureus* in each well are 6×10^4 cells and 1×10^7 CFU/mL, respectively. From the MTT assay results (Fig. R1a), we can see that HiZP has a negligible cytotoxicity against NIH/3T3 cells, demonstrating its good biocompatibility. However, the presence of *S. aureus* can seriously affect the survival of NIH/3T3 cells, and almost no surviving cells (< 3%) were observed after 50 h of incubation. Meanwhile, it was found that with the decrease of the viability of NIH/3T3 cells, the number of surviving *S. aureus* was remarkably increased in the co-culture system (upper figure of Fig. R1b). This is accord with the bacterial contamination effects to cell culture. However, upon the addition of HiZP into the co-culture system, a high viability (> 85%) can be observed in the NIH/3T3 cells, while *S. aureus* shows a decreased surviving rate and was completely killed after 40 h of incubation (lower figure of Fig. R1b). This indicates that the addition of HiZP can cause *S. aureus* inactivation to enhance the survival of NIH/3T3 cells, which is consistent with the time-dependent increase of ROS generation in the co-culture system of NIH/3T3 cells with *S. aureus* and HiZP, as reflected by the enhancement of DCF fluorescence in Fig. R1c. By contrast, no DCF fluorescence was recorded from the co-culture systems of NIH/3T3 cells with individual HiZP and *S. aureus*, revealing that no ROS was generated in these two systems. Apparently, these results demonstrated the feasibility of using HiZP as a prodrug system for on-demand bacterial inactivation and infection treatment with minimal side effects. We have added the results in the revised manuscript, please see new Figure S23.

Figure R1. (a) Viability of NIH/3T3 cells in the co-culture systems of NIH/3T3 cells with HiZP, *S. aureus*, and *S. aureus* and HiZP. (b) Viability of *S. aureus* in the co-culture systems of NIH/3T3 cells with *S. aureus* (upper) and *S. aureus* and HiZP (lower). (c) Time-dependent DCF fluorescence in the co-culture systems of NIH/3T3 cells with HiZP, *S. aureus*, and *S. aureus* and HiZP.

Q2. Authors responded that fluorescence microscopy image was taken without washing the biofilm deposition in PBS buffer. However, it is not clear how they prepared the biofilm. Whether the biofilm was formed on the multiwell plate and visualized under the fluorescence microscopy following staining with dyes or biofilm was formed on the glass slides and biofilm was imaged under the fluorescence microscopy following staining with dyes. Authors are suggested to clearly rewrite the experimental section.

Reply: The biofilm was formed on a 96-well plate and visualized under the fluorescence microscopy following staining with dyes. According to your suggestion, we have rewritten the experimental section of the preparation procedures of biofilm in the revised manuscript, please see p.21-22.

Q3. The membrane depolarization assay, measured by membrane potential indicator dye DiBAC4(3), is not mentioned in the experimental section.

Reply: We have added an experimental section of the membrane depolarization assay in the revised manuscript, please see p.22.

REVIEWERS' COMMENTS

Reviewer #4 (Remarks to the Author):

I have gone through the revised manuscript and found that authors satisfactorily addressed all the comments raised by this reviewer. The manuscript may be accepted for publication.

Responses to Reviewers' Comments

Reviewer #4 (Remarks to the Author):

I have gone through the revised manuscript and found that authors satisfactorily addressed all the comments raised by this reviewer. The manuscript may be accepted for publication.

Reply: We appreciate reviewer's positive comment.